# Role of Na^+^/Ca^2+^ Exchanger (NCX) in Glioblastoma Cell Migration (In Vitro)

**DOI:** 10.3390/ijms241612673

**Published:** 2023-08-11

**Authors:** Federico Brandalise, Martino Ramieri, Emanuela Pastorelli, Erica Cecilia Priori, Daniela Ratto, Maria Teresa Venuti, Elisa Roda, Francesca Talpo, Paola Rossi

**Affiliations:** 1Department of Biosciences, University of Milan, 20133 Milano, Italy; federico.brandalise@unimi.it; 2Department of Biology and Biotechnology “L. Spallanzani”, University of Pavia, 27100 Pavia, Italy; martino.ramieri01@universitadipavia.it (M.R.); emanuela.pastorelli@unipv.it (E.P.); ericacecilia.priori@unipv.it (E.C.P.); daniela.ratto@unipv.it (D.R.); mariateresa.venuti01@universitadipavia.it (M.T.V.); 3Laboratory of Clinical & Experimental Toxicology, Pavia Poison Centre, National Toxicology Information Centre, Toxicology Unit, Istituti Clinici Scientifici Maugeri IRCCS Pavia, 27100 Pavia, Italy; elisa.roda@icsmaugeri.it

**Keywords:** glioblastoma, NCX, BK channel, cell migration, brain tumor

## Abstract

Glioblastoma (GBM) is the most malignant form of primary brain tumor. It is characterized by the presence of highly invasive cancer cells infiltrating the brain by hijacking neuronal mechanisms and interacting with non-neuronal cell types, such as astrocytes and endothelial cells. To enter the interstitial space of the brain parenchyma, GBM cells significantly shrink their volume and extend the invadopodia and lamellipodia by modulating their membrane conductance repertoire. However, the changes in the compartment-specific ionic dynamics involved in this process are still not fully understood. Here, using noninvasive perforated patch-clamp and live imaging approaches on various GBM cell lines during a wound-healing assay, we demonstrate that the sodium-calcium exchanger (NCX) is highly expressed in the lamellipodia compartment, is functionally active during GBM cell migration, and correlates with the overexpression of large conductance K+ channel (BK) potassium channels. Furthermore, a NCX blockade impairs lamellipodia formation and maintenance, as well as GBM cell migration. In conclusion, the functional expression of the NCX in the lamellipodia of GBM cells at the migrating front is a conditio sine qua non for the invasion strategy of these malignant cells and thus represents a potential target for brain tumor treatment.

## 1. Introduction

Glioblastoma (GBM) is the most common and aggressive primary brain tumor [1]. Patients have a median survival rate of 14.6 to 16.7 months and a progression-free survival rate of 6.2 to 7.5 months [2,3]. Only 30% of GBM patients survive longer than one year after diagnosis [2]. GBM represents 30% to 60% of all the central nervous system (CNS) primary tumors [4]. Compared to other CNS cancers, GBM is characterized by a pronounced invasiveness that contributes to a poor prognosis [2]. GBM is resistant to the currently available therapies, including surgical resection accompanied by chemotherapy and radiotherapy [5]. Indeed, the aggressive invasiveness and infiltrative behavior of glioma cells precludes complete surgical resection, with a consequent rapid relapse [6].

Numerous genetic mutations have been found to be associated with GBM, affecting oncogenes and tumor suppressor genes [6]. Interestingly, mutations affecting ion channels and pumps have been found in 90% of GBM cases [7] and seem to be responsible for their highly proliferative rate, as well as for the migratory behavior and invasiveness of these tumoral cells [8,9]. For this reason, the term oncochannels has been coined to define the ion channels most commonly related to tumor development [8,9,10], which therefore constitute a potential pharmacological target for future treatment. Members of the Ca^2+^-activated K^+^ channel family, such as KCa3.1 (intermediate conductance K^+^ channel or IK) and KCa1.1 (large conductance K^+^ channel or BK), are especially involved in the development of gliomas by determining their uncontrolled growth, cell volume variation, and tumor cell migration [11]. An overexpression of BK channels has been reported in cell lines of GBM and in biopsies from GBM patients [12]. In addition, the BK channel structure is altered in GBM cells because of the expression of a splice variant producing the so-called gBK channel that displays an increased sensitivity to the intracellular calcium concentration ([Ca^2+^]i) [13].

The fine regulation of the intracellular Ca^2+^ and Na^+^ concentrations is a key physiological process. Intracellular Ca^2+^ works as a crucial signaling ion in both cytosolic and nuclear compartments, whereas Na^+^ plays an essential role in regulating cellular osmolarity and cytoskeletal remodeling [14,15]. In parallel with voltage-gated ion channels and ATP-dependent pumps, the Na^+^/Ca^2+^ exchanger (NCX) also contributes to maintaining the physiological concentrations of Na^+^ and Ca^2+^ by mediating their bidirectional flux [16]. NCX has been already subject to investigation in GBM—in particular, related to the activity of the TRPC channel [17] or to Ca^2+^-mediated cell death [18]. However, there is a paucity of reports on its role in GBM cell migration. The generally accepted stoichiometry of the NCX is an influx of three Na^+^ ions for the extrusion of one Ca^2+^ ion. However, depending on the intracellular concentrations of Na^+^ and Ca^2+^, it has been demonstrated that the NCX ionic flux ratio can vary from a minimum of 1:1 to a maximum of 4:1 [19,20]. When the intracellular Ca^2+^ concentration rises, the return of the Ca^2+^ concentration to the resting level is promoted by the forward mode of operation of the exchanger, coupling the influx of Na^+^ to the uphill extrusion of Ca^2+^. If the intracellular Na^+^ concentration rises or membrane depolarization occurs, the reduced transmembrane electrochemical gradient for Na^+^ leads the NCX to mediate the extrusion of intracellular Na^+^ and the influx of Ca^2+^, thus acting in reverse mode.

Three dominant genes comprising the NCX family have been identified in mammals, coding for the NCX1, NCX2, and NCX3 proteins. The NCX1 isoform is expressed in several tissues, including neurons and glial cells [21], whereas the NCX2 and NCX3 proteins have been found exclusively in neuronal and skeletal muscle.

Herein, we investigated the changes in the membrane conductance repertoire adopted by GBM cells during lamellipodia formation and their consequent migration in an in vitro model of invasion by using a battery of complementary assays and methods, e.g., the wound-healing assay, real-time time-lapse microscopy, immunofluorescence, and the noninvasive perforated patch-clamp technique. In particular, we focused on the functional expression of NCX in cells located within the migrating front, employing different GBM cell lines, i.e., U251 and U87, and a primary human glioblastoma culture enriched in stem cells (GSCs). Finally, we speculate about the osmotic and ionic mechanisms involving the NCX in promoting the formation of the migrating front during invasion.

## 2. Results

### 2.1. Cell Migration during a Wound-Healing Assay

We studied the behavior of U251 cells during 2D migration in a wound-healing scratch assay in monolayer confluent cultures, monitoring migration using real-time time-lapse microscopy throughout a 20 h time window. We recognized migrating cells based on their typical “polarized phenotype” characterized by digitiform and lamellipodia-like structures, oriented toward the empty area. We defined these cells as “edge cells”, meaning cells at the edge of wound healing (for a comparison of cell morphologies, see the inserts in Figure 1a). Through time-lapse microscopy, we observed that cells at the edge of the scratch migrated longitudinally towards the empty area, causing a significant increase in wound closure from 31.60% after 5 h to 88.47% after 15 h. The wound area produced by the scratch was, on average, 400 µm wide before the start of migration, and complete closure of the wound area was obtained 20 h after the scratch (Figure 1b). Subsequently, we analyzed the movements of these cells during migration, focusing on their position and trajectory at different time points. Cells at the edge of the scratch initially moved following a straight X-positive direction; then, after covering about 100 µm, they started to move in both the X-positive and Y-positive directions, toward the empty residual scratch area (Figure 1; see multimedia Appendix A). The dynamic of the migration process suggested that intercellular interactions during cell invasion played a significant role when the cells at the migrating front were at 100–200 µm from each other [22].

Cells at the edge moved with an average speed of about 0.6 µm/min, whereas cells located outside the wounded area moved slower, at an average speed of about 0.2 µm/min, and followed a random trajectory (Figure 1a,c,d). The speed difference between cells in the two scratch positions was highly statistically significant (extra-scratch: 0.18 ± 0.01 µm/min, n = 40; edge: 0.57 ± 0.02 µm/min, n = 40, *p*-value = 2.93 × 10^−30^; Figure 1e), indicating that cells at the edge of the wounded area were committed to migration, and their direction and speed were directed to fill the empty area. Furthermore, in the light microscopy phase contrast, these cells at the edge of the wounded area displayed a typical “polarized phenotype” with a digitiform structure oriented towards the empty area (insert in Figure 1a, 5 h).

The migrating front can be appreciated in the time-lapse microscopy video (see multimedia Appendix A).

### 2.2. Noninvasive Perforated Patch-Clamp Recordings Reveal Dramatic Changes in the Physiology of GBM Cells at the Edge of the Migrating Front

Noninvasive perforated patch-clamp recordings achieved by adding amphotericin in the patch pipette were chosen to avoid intracellular dialysis and changes in the physiological intracellular ions. Whole-cell perforated patch-clamp recordings were performed 1–4 h after the scratch procedure.

#### 2.2.1. Electrophysiological Passive Properties

Perforated patch-clamp recordings were performed on cells located in different positions of the scratch area: outside (extra, n = 25), inside (intra, n = 39), and at the edge of the scratch area (edge, n = 23; Figure 2a). GBM cells were classified as “edge cells” when, in a bright field, they showed a typical phenotype characterized by digitiform and lamellipodia-like structures polarized towards the wound healing but they did not yet enter inside the scratch (see Figure 1a). “Extra” cells were classified as cells that did not enter the scratch and did not have a lamellipodia-like structure (they usually had a typical bipolar morphology). “Intra cells” were classified as the ones that had already entered the wound healing.

The passive parameters were measured by analyzing the capacitive transient (see Section 4), and scattered values were reported (Figure 2).

The membrane potential (Vm), membrane capacitance (Cm), and membrane resistance (Rm) of the cells recorded within (intra, n = 39) and outside (extra, n = 25) the wounded area were significantly different (*p*-value = 0.043, 1.24 × 10^−7^, and 0.0012, respectively), in accordance with what has been reported previously [8] (Figure 2). In particular, the Vm was more depolarized, Cm higher, and Rm lower for the cells recorded within the scratch compared to those located outside [8] (Figure 2). As concerns the cells recorded at the edge of the wounded area taking part in the migrating front, the Vm value was not statistically different from those of the other two groups (Figure 2c). The Cm of cells recorded at the edge were similar to the values of cells recorded within the wounded area and about five times higher compared to the cells recorded outside (Figure 2d, *p*-value = 5.17 × 10^−10^). Furthermore, the Rm of the cells recorded at the edge of the wounded area were significantly lower than the Rm of the cells derived both within (*p*-value = 0.0001) and outside the wounded area (*p*-value = 9.04 × 10^−10^, Figure 2e). Therefore, the high Cm and low Rm correlated with a larger cell surface and a bigger membrane conductance in the cells committed to migration.

#### 2.2.2. Residual Inward Rectifier Potassium Current

Voltage-dependent currents were investigated in GBM cells recorded at different locations of the scratch area. Firstly, we used a hyperpolarizing voltage steps protocol where we applied consecutive −10 mV steps, from a holding potential of −40 mV down to −120 mV.

The cells inside the scratch displayed a residual inward rectifier current (Figure 3b) that was also present, although reduced in amplitude, in a fraction of the “edge” cells (n = 13/23; Figure 3c and Table 1). No current was recorded outside of the wounded area (Figure 3a and Table 1 [8]).

The residual inward rectifier current was blocked by 1 mM barium (Figure 3b,c, central panel), and its biophysical properties suggested the involvement of Kir4.1 channels [8].

In particular, the amplitudes of the inward current recorded at −120 mV were significantly higher in the cells inside the scratch compared to the other experimental conditions (Figure 3d; extra n = 25; intra n = 37; edge n = 19; extra vs. intra *p*-value = 3.02 × 10^−8^; extra vs. edge *p*-value = 0.33 (not significant); intra vs. edge *p*-value = 0.00026). In accordance, the inward current density recorded at −120 mV was also significantly higher in the cells inside compared to the other conditions (Figure 3e; extra n = 25; intra n = 37; edge n = 19; extra vs. intra *p*-value = 3.11 × 10^−8^; extra vs. edge *p*-value = 0.99 (not significant); intra vs. edge *p*-value = 0.0000195).

#### 2.2.3. “Edge” GBM Cells Expressed BK Channels and Expressed a Peculiar Inward Current

Voltage-dependent currents elicited by +10 mV depolarizing voltage steps from a holding potential of −60 mV up to +140 mV were subsequently investigated in GBM cells located at different locations of the scratch area.

Cells recorded outside the scratch (“extra”) exhibited a small outward current in response to consecutive depolarizing steps (Figure 4a; n = 25), while cells recorded within the wounded area showed a greater outward rectifier current that was TEA- and clotrimazole-sensitive, confirming previously published data [8,9] (Figure 4b,c and Table 1). Notably, cells recorded at the edge of the scratch displayed a larger outward rectifier current compared to the cells recorded both inside (*p*-value = 0.038) and outside (*p*-value = 1.23 × 10^−7^) the wounded area.

Surprisingly, during the depolarizing protocol, a constitutive inward current was recorded in the cells located at the edge of the wounded area (see the arrow in Figure 4c). In cells recorded at the edge, at test potentials more positive than +50 mV, the inward current was followed by an outward rectifier current (n = 18/20, Figure 4c, left traces). Therefore, to isolate the inward current from all the possible outward potassium currents, we locally perfused TEA and clotrimazole (Figure 4c, middle traces), blockers of both the BK and IK potassium channels. The experimental traces for the TEA- and clotrimazole-sensitive current, obtained by digital subtraction (Figure 4c, right traces), were used to quantify the outward currents. The amplitudes of the outward currents recorded at +140 mV were significantly different among the three groups of cells at different locations of the scratch area (Figure 4d; extra n = 25; intra n = 37; edge n = 23; extra vs. intra *p*-value = 0.00025; extra vs. edge *p*-value = 0.00000012; intra vs. edge *p*-value = 0.038). Accordingly, the outward current densities recorded at +140 mV were significantly different among the three groups of cells at different locations of the scratch area (Figure 4e; extra n = 25; intra n = 37; edge n = 23; extra vs. intra *p*-value = 0.00795; extra vs. edge *p*-value = 0.00000042; intra vs. edge *p*-value = 0.00819).

In addition, the averaged current–voltage relationship was computed for the currents recorded in cells at the edge of the scratch, both in the control conditions, and after TEA and clotrimazole local perfusion (Figure 4f). The inward current found in these cells displayed a peculiar current–voltage relationship, increasing proportionally to the depolarized test potentials. We further investigated this inward current to examine its biophysical and pharmacological properties.

#### 2.2.4. Electrophysiological Characterization of the Depolarization-Mediated Inward Current in “Edge” GBM Cells

To further characterize the unknown inward current recorded from migrating cells at the edge of the wound, we tested different depolarizing step protocols.

It should be noted that, at test potentials where the BK current activates, the total current recorded corresponds to the algebraic sum of the inward and the outward BK rectifier current. To reduce or avoid contamination of the BK current, and to enlarge the interval of potentials to be studied, we elicited the steps from –120 mV.

Furthermore, to verify whether the inward current was present in different cell lines, we repeated the same wound-healing assay on a human glioblastoma primary culture enriched in stem cells (GSCs) and on the U87 cell line. Cells were recorded at the edge of the wound healing and chosen following the above-mentioned criteria: the position at the edge, the “polarized phenotype”, and the presence of the digitiform structure, lamellipodia-like. After eliciting the same depolarizing protocol from −120 mV, 1–4 h after the scratch, an inward current was recorded in 87.15% (n = 14/16) of U251 cells, 83.3% of GSCs (n = 10/12), and 83% of U87 cells (n = 8/10) (Figure 5).

The similarity among the three cell lines, in terms of the current–voltage relationship and time to peak values (Figure 5, central and right panels), revealed that, in edge cells, all the lines expressed this depolarization-mediated inward current, suggesting that this current is a common feature in GBM migrating cells. Figure 5d shows the mean amplitude of the inward current recorded at +40 mV in U251 (n = 14), U87 (n = 8), and mesenchymal GBM (n = 10) edge cells. The amplitudes were not statistically different among the three tested cell lines.

Noteworthy is the consistency of the current–voltage relationship among the three tested cell lines (middle panels in Figure 5a–c), as well as the similarity in the time to peak values and voltage independence of this parameter (right panels in Figure 5a–c).

#### 2.2.5. The Inward Current of the Edge Cells Is Sensitive to a Reduction of Extracellular Na^+^ Consistent with Activation of the Sodium-Calcium Exchanger (NCX)

The inward current recorded from cells at the edge of the scratch showed a peculiar current–voltage relationship that did not correspond to a voltage-dependent current (Figure 5, middle panels). We hypothesized that this current could arise by the activation of specific pumps or exchangers involved in the complex modulation of the GBM migrating cells. Specifically, we hypothesized a crucial role for the sodium-calcium exchanger (NCX) in the generation of this peculiar current.

We therefore performed a set of experiments aimed at testing this idea by analyzing the effects of blocking the NCX in the inward current.

A “Na-free” extracellular solution was applied to block NCX activity [23]. The inward current amplitude was at first assessed under the control conditions in U251 edge cells; the subsequent perfusion of the “Na-free” solution (see Section 4) robustly inhibited the inward current (Figure 6a). On average, the amplitude recorded at +40 mV decreased from −368.14 ± 75.99 pA to −109.74 ± 59.56 pA (n = 5, *p*-value = 0.022, Figure 6b).

#### 2.2.6. Pharmacological Identification of the Sodium-Calcium Exchanger (NCX) and Its Role in the Migration Process

To confirm the involvement of the NCX, we performed an extensive pharmacological characterization of the inward current, administering drugs that specifically interfered with NCX activity. NCX is inhibited by the external perfusion of nickel [24,25]. To isolate the putative NCX-mediated component of the inward current, we recorded “edge” U251 cells before and after nickel local perfusion (1 mM). In this case, the recordings were performed in the presence of clotrimazole 10 µM and TEA 3 mM to be sure to isolate the sole inward current and then accurately isolate the nickel-sensitive component. The nickel application significantly reduced the inward current, as shown in Figure 7a. At +140 mV, the current was reduced from −222.39 ± 32.59 pA to −22.01 ± 53.06 pA (n = 5; *p*-value = 0.042; paired *t*-test, Figure 7b). The time constant of the inward current was consistent with the pool recorded in Figure 5 (n = 5; Figure 7c).

To further prove the NCX-mediated nature of the recorded inward current, we tested the effect of 5 min of the local perfusion of bepridil (100 µM), an antagonist of the NCX. Figure 8 illustrates experimental traces recorded in the control conditions and after the local perfusion of bepridil (100 µM). In the presence of bepridil, the average amplitude recorded at +40 mV decreased from −300.09 ± 85.95 pA to +34.19 ± 19.71 pA (n = 6, *p*-value = 0.049). The administration of the NCX antagonist almost abolished the inward current.

We also evaluated the effects of bepridil on the passive properties of the investigated cells. The membrane capacitance was significantly reduced when bepridil was added to the medium compared to the control conditions (control: 51.4 ± 4.6 pF, n = 5; bepridil: 40.9 ± 5.2 pF; *p*-value = 0.02), consistent with a reduction in the cell surface area. The membrane resistance was not significantly altered (control: 379.6 ± 14.0 MΩ, n = 5; bepridil: 163.1 ± 11.4 MΩ; *p*-value = 0.18).

#### 2.2.7. Effect of the Block of Na-K ATPase on the NCX Inward Current and the BK Outward Current

The NCX activity is strictly dependent on the intra- and extracellular Na^+^ and Ca^2+^ concentrations, whereas the BK current is sensitive to the intracellular Ca^2+^ concentration. We tested the effect of the Na-K ATPase blocker ouabain on the currents elicited by depolarization by applying two different protocols: (i) the first protocol started from a holding potential of −120 mV (Figure 9); (ii) the second protocol started from a holding potential of −60 mV (Figure 10). The former depolarizing protocol was chosen to study the NCX current and the latter to evaluate a possible effect of the Na-K ATPase blocker ouabain on the outward BK current [26].

Figure 9 shows experimental traces recorded in the control conditions and after the local perfusion of ouabain (500 µM) during the application of the first protocol starting from the −120 mV holding potential. Digital subtraction of the traces for the isolation of the ouabain-sensitive component is also displayed. After the local perfusion of ouabain, the amplitude at +40 mV decreased from −184.45 ± 30.29 pA to −59.31 ± 5.31 pA (n = 5, *p*-value = 0.018).

Because the block of the Na-K ATPase is associated with a reversed mode of the exchanger, which normally extrudes calcium from the cell [26], we tested a possible effect of changes in the Ca^2+^ concentration on the BK channels. To this end, we performed experiments using the second protocol, starting from the −60 mV holding potential, as shown in Figure 10. Following the perfusion of 500 µM ouabain, the I–V relationship shifted from inward to outward at the most hyperpolarized steps to +80 mV (see also the insert in Figure 10), and the amplitude of the outward current at +140 mv changed from 846.95 ± 282.79 pA to 1676.6 ± 629.57 pA (n = 5, *p*-value = 0.11, not significant) (Figure 10). The insert in Figure 10 shows the averaged I–V magnified in the interval range between −40 mV and +40 mV to make more visible the I–V shift from inward to outward after the local perfusion of ouabain.

Therefore, these data suggest that a reduction in the forward mode of the NCX and the decrease in the Ca^2+^ extrusion may indeed positively activate the BK and gBK currents. We cannot exclude that, for longer perfusion times, a change in ion gradients could switch the exchanger to the reverse mode.

### 2.3. Immunolabeling for NCX in the Lamellipodia

To clarify the role of the sodium-calcium exchanger (NCX) in the migration and invasiveness of U251 cells, we investigated the expression/localization of NCX in the wound-healing scratch assay with immunofluorescence using a primary monoclonal antibody for the NCX1 isoform (SLC8A1). We assessed the distribution of NCX1 immunopositivity in both the whole cytoplasm, as well as in the lamellipodia along the migrating front.

As regards the cytoplasmic area, in the control conditions (CTR, i.e., in the absence of the scratch), NCX1 immunolabeling was homogeneously distributed in the whole cytoplasm, with no specific localization (Figure 11a). After the scratch, a similar homogeneous immunolabeling pattern in NCX1 immunopositivity was observed, with no significant change (Figure 11a). Notably, NCX1 immunofluorescent labeling was particularly intense at all the evaluated time points, without any statistically significant optical density (OD) difference in time (OD ranged between 14.1 and 15.1) (Figure 11b). These findings clearly support the notion that NCX cytoplasmic expression is not affected by the scratch.

In contrast, concerning the lamellipodia formations at the migration front, NCX1 immunolabeling was increasingly enhanced over time, becoming particularly marked at later time points on the leading border of the edge cells (Figure 11a). Interestingly, a significant difference in OD values was noted already 1 h after the scratch compared to the CTR. Further, a significant diversity of the OD values was assessed when comparing different time points in the same cells (Figure 11c,d).

### 2.4. Blockade of NCX Significantly Reduced U251 Cell Migration in the Wound-Healing Assay

To investigate the role of the NCX in promoting the migration of GBM cells, we performed the wound-healing assay on U251 cells treated with nickel and bepridil, and we monitored their migration using time-lapse microscopy.

We compared the migration of U251 cells during the healing scratch assay in the presence of nickel (200 µM; Appendix A) by tracking the cells located at the edge (“edge”) and those situated outside the scratch (“extra”) (Figure 12a). In the presence of nickel, cells migrated with a random not finalized Y-positive trajectory from the beginning, resulting in reduced scratch closure compared to the control conditions (Figure 12b–d). The migration speed of the cells located both outside the scratch and at the edge in the nickel condition was slower compared to the control conditions. In particular, for “extra” cells, the migration speed was 50% lower than in the control conditions (extra: 0.09 ± 0.01 µm/min in nickel vs. extra control 0.18 ± 0.01 µm/min, n = 40 for both experimental conditions, *p*-value = 1.14 × 10^−9^, unpaired *t*-test, Figure 12d). For “edge” cells, the migration speed was about 30% slower compared to the control (edge: 0.39 ± 0.01 μm/min in nickel vs. edge control 0.57 ± 0.02 µm/min, n = 40 for both experimental conditions, *p*-value = 9.11 × 10^−14^, unpaired *t*-test). The difference in values between “extra” and “edge” cells in the presence of nickel was statistically significant (*p*-value = 9.07 × 10^−43^, unpaired *t*-test; Figure 12e).

We then replicated the same test using bepridil to functionally block the NCX. At first, to exclude its possible cytotoxic effect, we assessed cell viability with the MTT assay using increasing concentrations (20, 50, 70, and 100 μM; see the Section 4) of bepridil for 24 h. Our findings revealed that bepridil at 20 μM did not affect cell viability, with almost all cells surviving. However, at 50 μM, cell viability was reduced by about 30%. The half-maximal inhibition (IC50) of cell viability was at about 60 μM. We therefore chose a concentration of 50 μM, leading to a mild reduction in the number of living/proliferating cells, for the time-lapse microscopy analysis (Figure 13). In the presence of bepridil, U251 cell migration followed a peculiar trajectory (Figure 14 and Appendix A). Similar to the nickel condition, slower migration was detected for both the cells located outside the scratch and at the edge compared to the related control conditions (extra: 0.05 ± 0.01 µm/min in bepridil vs. extra control 0.18 ± 0.01 µm/min, n = 40 for both experimental conditions, *p*-value = 2.00 × 10^−18^; edge: 0.25 ± 0.02 μm/min in bepridil vs. edge control 0.57 ± 0.02 µm/min, n = 40 for both experimental conditions, *p*-value = 1.36 × 10^−22^; unpaired *t*-test; Figure 14d). The difference in values between “extra” and “edge” cells in the presence of bepridil was statistically significant (*p*-value = 4.13 × 10^−20^; unpaired *t*-test; Figure 14e).

## 3. Discussion

Glioblastoma is characterized by its unique ability to grow in a confined space and to infiltrate the cerebral parenchyma. The characterization of ion channels, exchangers, and transporters contributing to the devastating invasiveness of GBM is a crucial task that requires further investigation.

Available data describing ion channels, transporters, and exchangers in GBM hypothesize two main processes to explain their role in cell invasion: (i) the formation of a hyperosmotic intracellular environment that triggers a water influx with a consequent cellular volume increase at the leading edge of the cells; (ii) cell shrinkage of the posterior edge of the cell achieved by an efflux of salt, followed by a reduction in the cytosolic water content [11,14,27,28]. In particular, the current assumptions suggest that, through NKCCl and Na-K ATPase, Cl^-^ and K^+^ both accumulated intracellularly, giving rise to the electrochemical driving force needed for triggering these osmotic effects [11,14,27,28]. More recently, other conductance has been described to play a major role in the regulation of the chloride dynamics, such as the volume-regulated anion channel (VRAC/LRRC8) [29,30]. However, the importance of the VRAC/LRRC8 channel in GBM migration is still under debate [31] due to possible variabilities in both the cell line adopted between these works and the migration model used.

One of the strategies adopted by tumor cells to improve proliferation and survival is the up- and downregulation of intracellular Ca^2+^ levels, thus gaining control of intracellular signaling and the metabolism [32]. To this aim, the expression, function, and localization of Ca^2+^ pumps, channels, and exchangers are strongly modulated in cancer cells, although the molecular pathways recruited for these detrimental alterations are still poorly understood, and further research is needed.

HereIn the current study we demonstrated, for the first time, the involvement of the sodium-calcium exchanger (NCX) in GBM-invasive processes, and we tried to elucidate the possible mechanisms responsible for its alteration.

NCX is ubiquitously expressed in mammals, where it exists in three isoforms (NCX1, NCX2, and NCX3), distributed in the plasma membrane. NCX plays a major role in the homeostasis of the intracellular Ca^2+^ concentration, presenting two distinct functional modes: in the so-called forward mode, NCX provides an effective influx of three Na^+^ ions to the uphill extrusion of one Ca^2+^, whereas, in the reverse mode, it promotes a Ca^2+^ influx against the Na^+^ efflux. Intriguingly it has also been shown to be involved in the regulation of the migration of the microglia [33]. A large number of effectors can modulate NCX expression and activity, from the pH and hypoxia to intracellular phosphorylation [34,35]. This might explain the unexpected current–voltage relationship of the exchanger at high depolarizing values described here, which we intend to examine further in future investigations.

Several studies have shown the involvement of NCX in cardiac disease, and a therapeutic potential has been proposed for NCX inhibitors to treat heart injuries [36]. However, the experimental evidence for the functional role of NCX in glioblastoma has not been fully clarified [37]. To date, NCX has been reported to play a role mainly in proliferation and tumor growth [18,38], but little is known about its role in GBM cell migration. In prostate cancer, melanoma, and leukemia, NCX preferentially works in the reverse mode, although the lack of specificity of the KB-R7943 inhibitor used in these investigations complicates the interpretation of the results [35,37].

Our data collected with time-lapse microscopy applied during the wound-healing assay identified, at the edge of the scratch, cells committed to migration. These cells displayed a characteristic feature, the lamellipodia, and were capable of migrating in a specific direction and at high speed to fill in the empty space created by the scratch. We recorded whole-cell currents from these cells by means of amphotericin-perforated patch-clamp so as to avoid intracellular dialysis and alterations of the intracellular calcium and chloride concentrations, as previously described [13].

We found that the cells committed to migration displayed a larger surface and a higher conductance compared to the nonmigrating cells, as demonstrated by the higher Cm and lower Rm, respectively. Upon the depolarizing steps, U251 cells showed a constitutive outward BK current with the same biophysical and pharmacological properties previously described [8,9]. The BK current recorded at the edge was significantly greater compared to the current recorded within the wounded area.

Unexpectedly, 90% of the edge cells displayed, in addition to the BK current, a constitutive inward current never recorded before, neither outside nor inside the wounded area. This peculiar inward current was activated during depolarization, started to appear at a membrane potential of about −20 mV, and displayed an “ohmic” I–V relationship at all tested membrane voltages (up to +140 mV) without presenting a time-dependent inactivation. The same inward current, with a similar amplitude, current–voltage relationship, and time to peak, was also present in GBM cells derived from human glioblastoma cultures enriched in mesenchymal stem cells (GSCs) and in U87 cells when recorded at the edge of the wound-healing assay in a perforated patch-clamp configuration. Therefore, we hypothesized that this inward current is a common feature of GBM.

Based on our pharmacological characterization of this peculiar inward current, we concluded it was mainly mediated by the NCX. Indeed, we demonstrated that this inward current was significantly reduced when recordings were performed in a sodium-free solution. The inward current was also sensitive to bepridil, an antagonist of the NCX. Since the NCX is inhibited by the external perfusion of nickel, we managed to isolate the NCX-mediated component (i.e., the nickel-sensitive component). To avoid potential incomplete blocks of the NCX, future experiments will also replicate our findings using a CRISPR/Cas9 KO model for the same cell lines. A large number of effectors can modulate the NCX expression and activity, from the pH and hypoxia to intracellular phosphorylation [34,35]. This might be one of the reasons why the current–voltage relationship of the exchanger at highly depolarized values has an unexpected trend. Additionally, it is possible that, at higher depolarized holding potentials, other conductance is also activated along with the NCX-mediated current. Further experiments are planned to investigate in greater detail the modulatory mechanisms of the NCX occurring during the invasion by GBM cells. To further confirm the role of NCX in glioblastoma, in vivo experiments should also be considered as in 2D cultures, and by using the scratch assay, we might have a relatively limited predictive value of glioblastoma cell behavior in a more complex and physiological environment.

We then confirmed the presence of NCX by immunofluorescence staining, showing that, during migration, the expression levels of this exchanger increased over time in the lamellipodia of the edge cells but not in the cytoplasm. Finally, a blockade of the forward Na^+^/Ca^2+^ exchanger by nickel and bepridil significantly slowed the time for wound closure [18], confirming an active role for the NCX in the GBM invasion process. All features of the migration process in the control conditions were impaired in the presence of nickel or bepridil, and the cells at the edge no longer exhibited the typical lamellipodial structures. Interestingly, in the presence of bepridil, the cells at the edge tended to emit a long “neurite-like structure”, and the soma was unable to cover a distance greater than 100 µm while exhibiting a very low speed during the migration process.

Subsequently, we demonstrated that NCX activity depended on a Na-K ATPase for the maintenance of the sodium gradient. The reduction in NCX activity and the subsequent increase in intracellular Ca^2+^ may be responsible for the increase in the BK/gBK currents. GBM cells can adopt other strategies to increase cytosolic calcium during migration, such as the opening of mechanosensitive potassium channels through hypotonicity, as reported in [39] (see also [28]). In conclusion, we propose that the Na^+^/Ca^2+^ exchanger could constitute a potential target in antitumor chemotherapy. The inhibition of the forward mode or the stimulation of the reverse mode of the NCX may result in increased intracellular Ca^2+^ concentrations, which could, in turn, trigger intracellular mechanisms of cell death in tumor cells. It is thus interesting to consider the possibility of modulating the aberrant activity of normal or altered NCX isoforms, selectively expressed in cancer cells, to disrupt intracellular Ca^2+^ homeostasis and inhibit GBM progression.

## 4. Materials and Methods

### 4.1. Cell Cultures

#### 4.1.1. Human U251 GBM Cell Line

Human U251 GBM cells (Sigma-Aldrich, Milan, Italy) were cultured in Eagle’s minimal essential medium (EMEM) supplemented with 10% fetal bovine serum (FBS), 1% L-glutamine, 1% non-essential amino acid (NEAA), 1% sodium pyruvate, and 1% penicillin–streptomycin. The cell culture was maintained in 25 or 75 cm^2^ flasks at 37 °C in a humidified atmosphere (95% air and 5% CO_2_). All culturing reagents were acquired from Euroclone S.p.a. (Pero, Milan, Italy).

#### 4.1.2. U87 Cell Line

Human U87 GBM cells (Sigma-Aldrich, Milan, Italy) were cultured in Dulbecco’s modified Eagle’s medium (DMEM) high glucose supplemented with 1% L-glutamine, 10% fetal bovine serum (FBS), and 1% penicillin–streptomycin (pen–strep) (Thermo Fisher Scientific, Monza, Italy). The cell culture was maintained in 25 or 75 cm^2^ flasks at 37 °C in a humidified atmosphere (95% air and 5% CO_2_). All culturing reagents were acquired from Euroclone S.p.a. (Pero, Milan, Italy).

#### 4.1.3. Human GBM CSC Cultures

The human glioblastoma culture enriched in stem cells (GSCs) was kindly provided by Professor T. Florio’s laboratory at the University of Genova (Genova, Italy). Briefly, the culture was obtained from biopsy specimens from patients who underwent surgical treatment, in the absence of previous therapies, at the Neurosurgery Department IRCCS-AOU San Martino-IST (Genova, Italy) after obtaining informed consent and approval from the Institutional Ethical Committee (IEC). Based on histological and immunohistochemical analyses, the samples were classified as grade IV GBM (referring to the WHO classification) [40].

GSCs were grown in a stem cell-permissive medium, including 1:1 DMEM high glucose and Ham’s F-12 Nutrient Mix GlutaMAX™ (Gibco^TM^, Thermo Fisher Scientific, Rodano, MI, Italy) enriched with B27 supplement (Gibco^TM^, Thermo Fisher Scientific, Rodano, MI, Italy), 10 ng/mL human basic fibroblast growth factor (bFGF), and 20 ng/mL human epidermal growth factor (EGF) (Miltenyi Biotec, Bologna, Italy). Cells were cultured as a monolayer on Matrigel (BD Biosciences, Milan, Italy), as previously described [40].

### 4.2. Cell Viability and Bepridil Cytotoxicity Determination Using the MTT Assay

To determine the optimal concentration of the NCX blocker bepridil (Sigma-Aldrich, Milan, Italy) to be used in the wound-healing assay and time-lapse microscopy, U251 cell viability was determined using the MTT [3-(4,5-dimethylthiazol-2-yl)-2,5-diphenyltetrazolium bromide] assay. Briefly, cells were cultured in a 25 cm^2^ flask, seeded in a 96-well plate at a density of 10,000 cells/well (0.2 mL per well), and incubated at 37 °C for 24 h in a humidified atmosphere containing 5% of CO_2_. The day after, the culture medium was replaced with fresh medium added with a range of bepridil concentrations (20, 50, 70, and 100 μM). For the control condition, cells were incubated with the culture medium alone. Twenty-four hours after exposure, the MTT solution (20 μL/well) was added to each well; this operation was performed in darkness, and the plates were subsequently incubated for about three hours at 37 °C. At the end of the incubation time, the quantification was performed by measuring the samples’ absorbance at 550 nm with the ELx808TM Absorbance Microplate Reader (Bio-Tek Instruments, Inc., Winooski, VT, USA). Data were expressed as a percentage of the control, and the cell viability percentage was calculated.

### 4.3. Wound-Healing Migration Assay and Time-Lapse Microscopy

U251, U87, and human GBM CSC cells were seeded on glass coverslips (22 mm × 22 mm) and in a 6-well plate for the wound-healing assay and time-lapse microscopy, respectively. Cells were grown in a complete medium and cultured until 90–95% confluence. A disposable pipette tip (1 mL volume) was used to scratch wounds on the midline of the coverslip (mean width: 350 μm), which was carefully washed with the complete medium to remove floating cells and clean the wound area.

Time-lapse imaging experiments were performed on the U251 cell line. The percentage of scratch wound closure was monitored using an Olympus ScanR inverted microscope equipped with a 10× objective taking pictures every 5 min over 24 h. The entire assay was performed using an environmental microscope incubator mimicking the culture conditions (37 °C and 5% CO_2_). The speed of scratch closure was calculated with LAS X—Leica Microsystems CMS GmbH Software (Version 5.1.0). In addition, individual cell motility and the percentage of area covered by the cells were analyzed in ImageJ using the Chemotaxis Tool ImageJ software plugin (Version 2.0). Experiments were performed at least in triplicate. Only cells that were present in the field of view (FOV) for the whole recording time were included in the analysis of the time lapse for all the conditions applied. Usually, cells located on the bottom left corner of the FOV and with an upward direction during the migration were selected and tracked. For the bepridil and nickel datasets, recording began right after the blocker was applied in the cell medium.

### 4.4. Electrophysiological Recording

Patch-clamp experiments were performed at room temperature as previously described [8]. Briefly, whole-cell perforated patch-clamp recordings were performed using an Axopatch-200B (Molecolar Devices, San Jose, CA, USA) patch-clamp amplifier at the output cut-off frequency of 5 kHz. The extracellular solution contained 140 mM NaCl, 5 mM KCl, 10 mM HEPES, 10 mM glucose, 3 mM CaCl_2_, and 1.2 mM MgSO_4_, with a pH of 7.4. For the “Na-free” experiment, the extracellular solution contained 140 mM LiCl, 5 mM KCl, 10 mM HEPES, 10 mM glucose, 3 mM CaCl_2_, and 1.2 mM MgSO_4_, with a pH of 7.4. Patch pipettes were filled with an intracellular solution containing 400 μg/mL amphotericin (Sigma-Aldrich, MI, Italy), 140 mM KCl, 4 mM NaCl, 5 mM EGTA, 5 mM HEPES, and 0.5 mM CaCl_2_, with a pH of 7.3. The series resistance (Rs = 16 ± 1.3 MΩ), input resistance (Rm), and membrane capacitance (Cm, Figure 2) were measured as previously described by eliciting a hyperpolarized step of −10 mV from the holding potential of −60 mV [41,42]. The whole-cell perforated patch-clamp stability was assessed every 5 min by monitoring the Rs. A depolarizing ramp from −120 to +140 mV was stimulated at a rate of 0.077 mV/ms and at a sampling rate of 500 Hz. Other protocols were reported in the figures. The hyperpolarizing protocol from −40 to −120 mV was elicited in steps of 10 mV at a sampling rate of 6.67 kHz, and the depolarizing protocols from –60 to +120 mV or from −120 to +40 mV were elicited in steps of 20 mV at a sampling rate of 20 kHz. For the depolarizing protocols, the PN leak subtraction of the Clampex program was used to eliminate the effects of the leakage current on the whole-cell responses.

The junction potential between the patch pipette and the cytoplasm was +8.8 mV, as calculated according to Horn [43], and the current–voltage (I–V) relationships were shifted to the left by this value.

The following compounds were used in the patch-clamp experiments: clotrimazole 10 µM, TEA 3 mM, barium 1 mM, nickel 1 mM, bepridil 100 µM, and ouabain 500 µM. Clotrimazole (5 mM), bepridil (100 µM), and ouabain (500 µM), as well as amphotericin, were dissolved in DMSO. All reagents were purchased from Sigma-Aldrich (Milan, Italy), except for bepridil, which was bought from Adooq Bioscience (Irvine, CA, USA).

### 4.5. Immunofluorescence Evaluations

The control and treated U251 cells were grown on coverslips (22 mm × 22 mm) placed in cell culture dishes (35 mm × 10 mm) until 90% confluence, fixed with 4% formalin (20 min), and post-fixed with 70% ethanol at −20 °C for at least 24 h.

The samples were rehydrated for 10 min in PBS and then incubated with 1% BSA to block nonspecific binding sites. Subsequently, the cells were immunolabeled using the anti-NCX1 isoform primary antibody (#MA3-926, mouse monoclonal anti-sodium-calcium exchanger SLC8A1, Thermo Fisher Scientific, Monza, Italy) diluted 1:200 in PBS (Sigma-Aldrich, Milan, Italy) for 1 h at RT in a dark, moist chamber. The specificity of the antibody has been extensively described in different models [44,45,46]. Then, the cells were washed with PBS and incubated for 45 min with the goat anti-mouse IgG (H + L) highly cross-adsorbed secondary antibody, Alexa Fluor Plus 488, diluted 1:200 in PBS (Thermo Fisher Scientific, Monza, Italy). DNA counterstaining was performed using 0.1 mg/mL of Hoechst 33258 (Sigma-Aldrich, Milan, Italy). Lastly, the cells were mounted with a drop of Mowiol (Calbiochem-Inalco, Italy) for microscopy visualization. Images were recorded using an Olympus BX51 microscope equipped with an Olympus MagniFire camera system and processed with Olympus Cell F and ImageJ software (Version 1.51) to analyze the mean optical density (OD) (ratio of the mean of immunofluorescence intensity on the cell surface). For each condition, 11 quadrants (about 50 cells) were evaluated for random analysis. The channels of each fluorescence were split to obtain single images in grayscale, where the minimum value was 0 (black) and the maximum value was 255 (white). Then, the OD values were normalized to the lamellipodia and cytoplasm areas.

### 4.6. Statistical Analysis

All data were reported as the mean ± standard error of the mean (SEM). The Bartlett and Shapiro–Wilk tests were performed to determine the normal distribution of the datasets. Statistical significance was determined using the *t*-test when comparing two experimental groups and one-way analysis of variance (ANOVA) with the post hoc Bonferroni test when comparing more than two experimental groups. *p*-values were considered statistically significant at * *p* < 0.05, ** *p* < 0.01, *** *p* < 0.001, and **** *p* < 0.0001. All statistical analyses were performed with GraphPad Prism 8.0 software (GraphPad Software Inc., La Jolla, CA, USA).

## Figures and Tables

**Figure 1 ijms-24-12673-f001:**
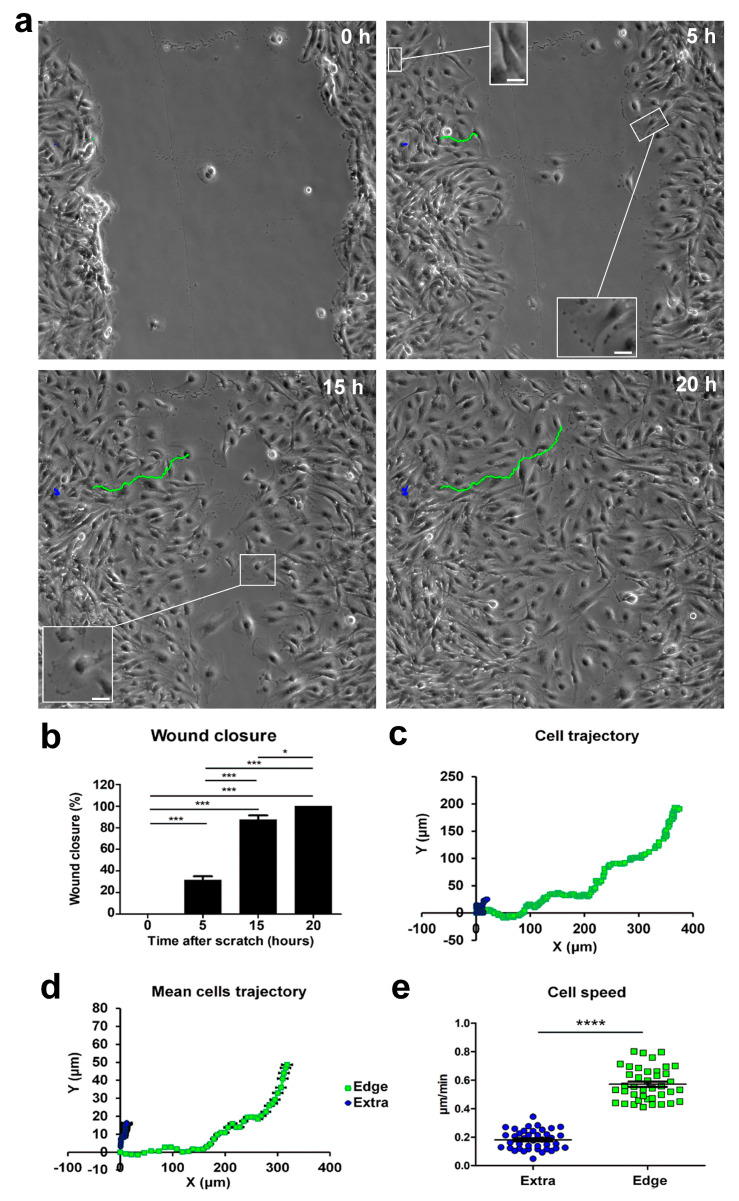
Time-lapse microscopy of cell migration in the wound-healing assay on U251 GBM cells. (**a**) Tracking of individual cells located in the extra-scratch (blue) and at the edge of the wounded area (green). Image shots at 0, 5, 15, and 20 h after the wound scratch. Scale bar: 100 µm. Inserts presenting high-magnification images of cells in different positions of the scratch and also showing a lamellipodia as a typical feature of the cells localized at the edge of the scratch. Scale bar: 10 µm (**b**) Wound closure percentages at 0, 5, 15, and 20 h after the wound scratch. (**c**) Plotting of the trajectories of one individual cell located outside (extra, blue) and one individual cell at the edge of the scratch (edge, green). Images were analyzed using the Chemotaxis Tool ImageJ software plugin (Version 2.0). (**d**) Mean trajectories of the cells outside (extra, blue, n = 40 cells) and at the edge of the scratch (edge, green, n = 40 cells). (**e**) Scatter plot of the velocity (µm/min) of single cells outside (extra, blue, n = 40 cells) and at the edge of the scratch (edge, green, n = 40 cells). *p*-values: * *p* < 0.05, *** *p* < 0.001, and **** *p* < 0.0001.

**Figure 2 ijms-24-12673-f002:**
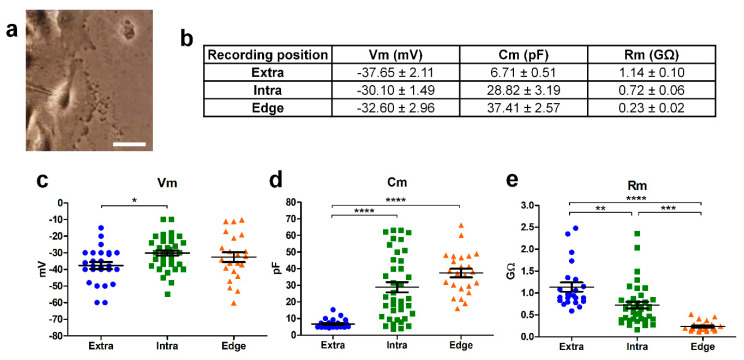
Passive parameters of the cells recorded outside (“Extra”; blue dots), within (“Intra”; green squares), and at the edge (orange triangles) of the wound area. (**a**) Example of a patch pipette approaching a cell at the edge of the scratch. Scale bar: 10 µm. (**b**) Resting membrane potential (Vm), membrane capacitance (Cm), and membrane resistance (Rm) mean values of cells recorded at different positions of the scratch. Scatter plots of the resting membrane potential (Vm; (**c**)), membrane capacitance (Cm; (**d**)), and membrane resistance (Rm; (**e**)). Number of cells: outside (n = 25), inside (n = 39), and at the edge of the scratch area (n = 23). *p*-values: * *p* < 0.05, ** *p* < 0.01, *** *p* < 0.001, and **** *p* < 0.0001.

**Figure 3 ijms-24-12673-f003:**
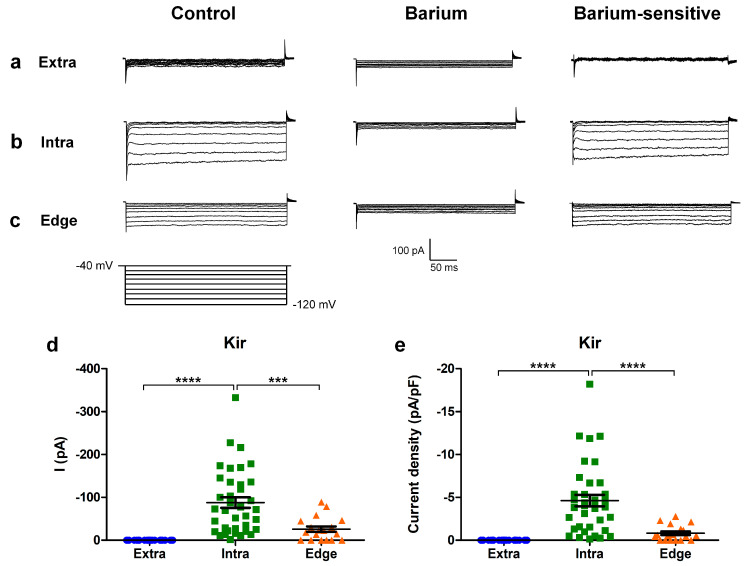
Hyperpolarizing protocol for investigating the inward rectifier current. (**a**,**b**) Experimental traces obtained using a hyperpolarizing voltage step protocol (bottom) in the cells recorded extra-scratch (**a**), intra-scratch (**b**), and at the edge (**c**) of the wounded area in the control conditions (left) and after perfusion with barium 1 mM (center). The barium-sensitive current was obtained by digital subtraction (right). (**d**) Scatter plot of the amplitude of the K_IR_ current measured at −120 mV from individual cells at different locations of the scratch area. (**e**) Scatter plot of the current density of the K_IR_ current measured at −120 mV from individual cells at different locations of the scratch area. *p*-values: *** *p* < 0.001, and **** *p* < 0.0001.

**Figure 4 ijms-24-12673-f004:**
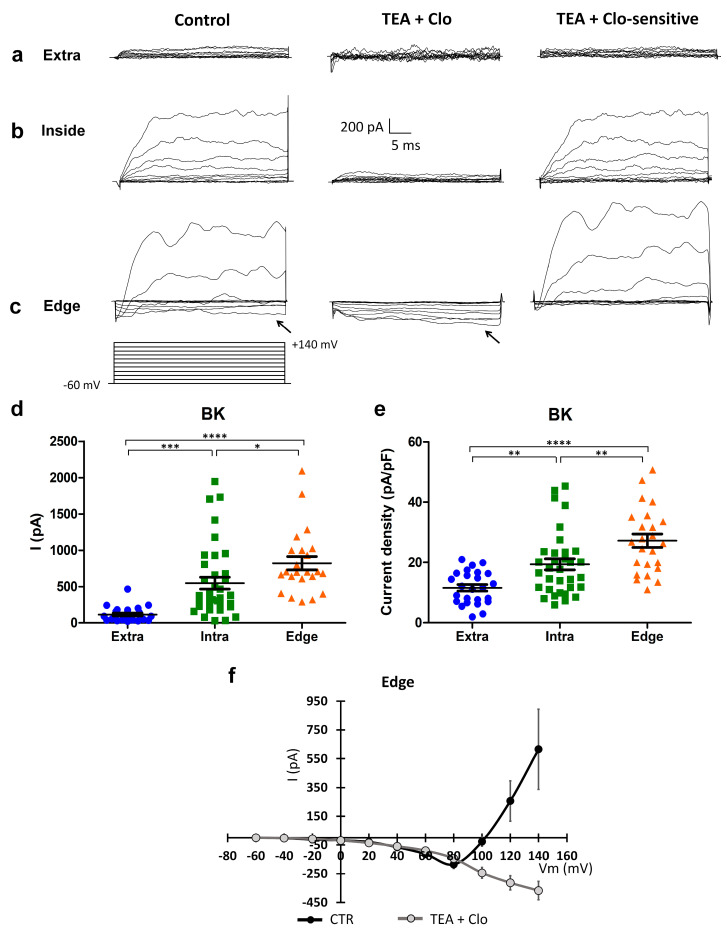
Expression of BK channels, and expression of a peculiar inward current elicited by a depolarization protocol in the “edge” GBM cells. (**a**–**c**) Currents elicited by depolarizing steps, from the holding potential of −60 mV, in cells recorded outside (**a**), within (**b**), and at the edge (**c**) of the wounded area in the control condition (left traces) and after clotrimazole 10 µM and TEA 3 mM local perfusion (central traces). Digital subtraction shows TEA- and clotrimazole-sensitive currents (right traces). The depolarizing voltage step protocol is shown at the bottom. Arrows indicate the inward current present in the control condition and after TEA and clotrimazole local perfusion in the cells recorded at the edge. (**d**) Scatter plot of single cell values showing the absolute values of BK currents at +140 mV recorded outside (extra, n = 25, blue dots), within (intra, n = 37, green squares), and at the edge (edge, n = 23, orange triangles) of the wounded area. (**e**) Scatter plot of single cell values showing the BK current density at +140 mV recorded outside (extra, n = 25, blue dots), within (intra, n = 37, green squares), and at the edge (edge, n = 23, orange triangles) of the wounded area. (**f**) Mean I–V relationship in the cells (n = 8) recorded at the edge of the wounded area in the control conditions (CTR) and after the local perfusion of TEA + clotrimazole. *p*-values: * *p* < 0.05, ** *p* < 0.01, *** *p* < 0.001, and **** *p* < 0.0001.

**Figure 5 ijms-24-12673-f005:**
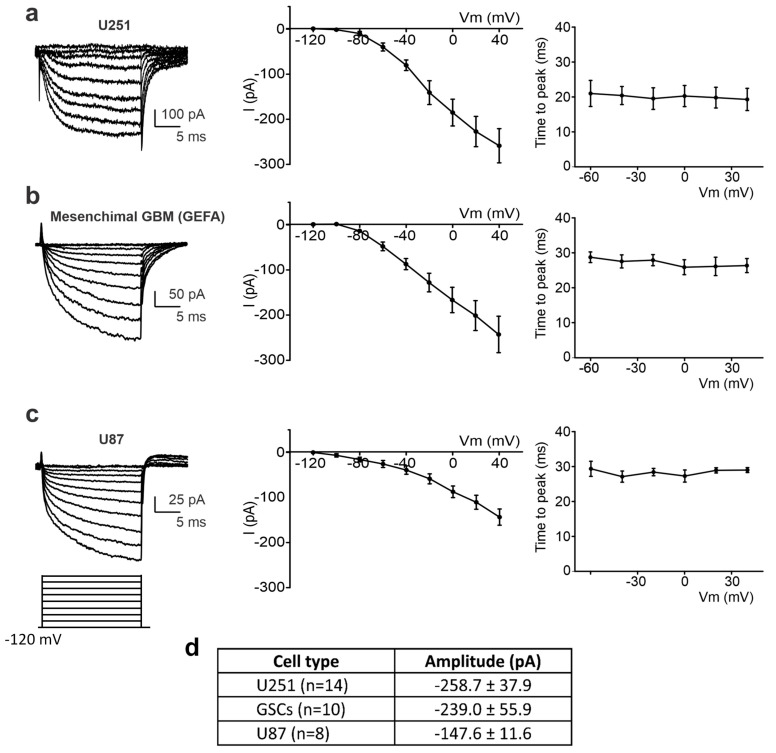
Detection of the inward current at the edge of the wounded area in different cell lines. (**a**–**c**) Left: experimental traces elicited by the application of a depolarizing protocol of the voltage steps (shown at the bottom) from a holding potential of −120 mV to + 40 mV in U251 (**a**), mesenchymal GSCs (**b**), and U87 cells (**c**). Center: mean I–V relationship for U251 (n = 14 cells, (**a**)), mesenchymal GSCs (n = 10 cells, (**b**)), and U87 (n = 8 cells, (**c**)). Right: average inward current activation time to peak measured at different test potentials for U251 (n = 14 cells, (**a**)), mesenchymal GSCs (n = 10 cells, (**b**)), and U87 (n = 8 cells, (**c**)). (**d**) Averaged amplitudes of the inward current at +40 mV, measured by means of a depolarizing step from a holding potential of −120 mV, recorded in different cell lines at the edge of the wounded area.

**Figure 6 ijms-24-12673-f006:**
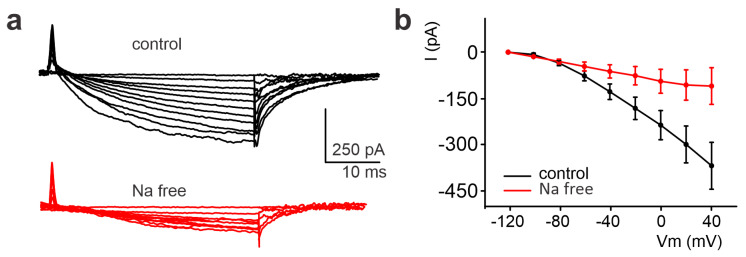
Significant reduction of the inward current detected in the U251 edge cells following the perfusion of a Na-free solution. (**a**) Experimental traces elicited by depolarizing the voltage steps from −120 mV to +40 mV before (black traces) and after local perfusion of the extracellular Na-free solution containing lithium chloride (LiCl, red traces) instead of sodium chloride (NaCl). (**b**) Mean I–V relationships before and after LiCl local perfusion (n = 5).

**Figure 7 ijms-24-12673-f007:**
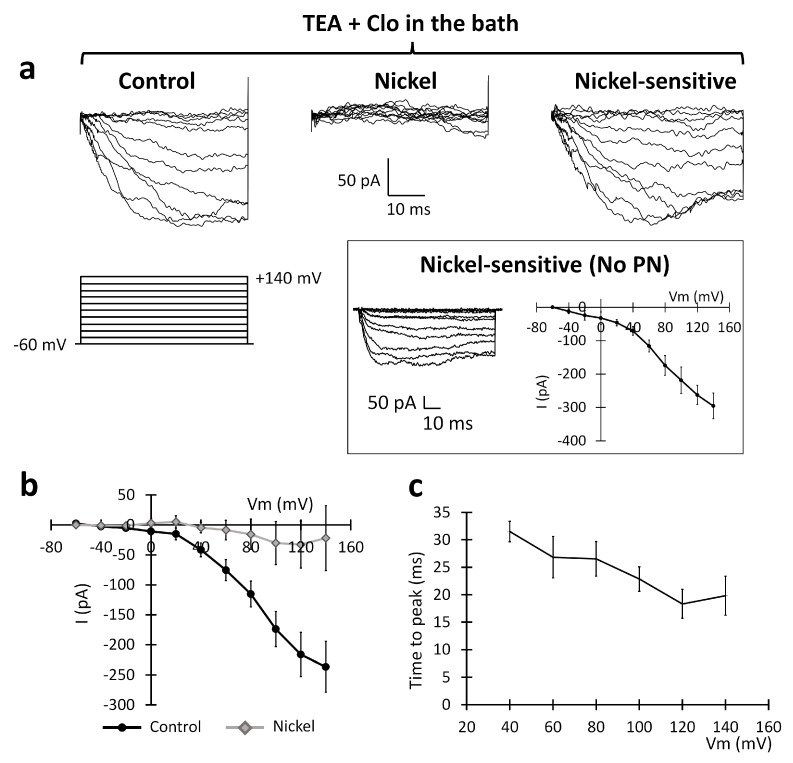
Sensitivity of the inward current recorded in the U251 edge cells to 1 mM nickel extracellular perfusion. (**a**) Experimental traces elicited by depolarizing the voltage steps of +10 mV from the holding potential of −60 mV to +140 mV (protocol shown at the bottom) in the control conditions (left traces) and after the local perfusion of 1 mM nickel (central traces). Digital subtraction shows the nickel-sensitive current (right traces). Inset: The PN leak subtraction method (see the Section 4) does not significantly alter the I–V plot of the inward current. (**b**) Mean I–V relationships before (black dots) and after (gray rhombuses) the local perfusion of nickel (n = 5 cells). (**c**) Average inward current activation time to peaks measured at different test potentials.

**Figure 8 ijms-24-12673-f008:**
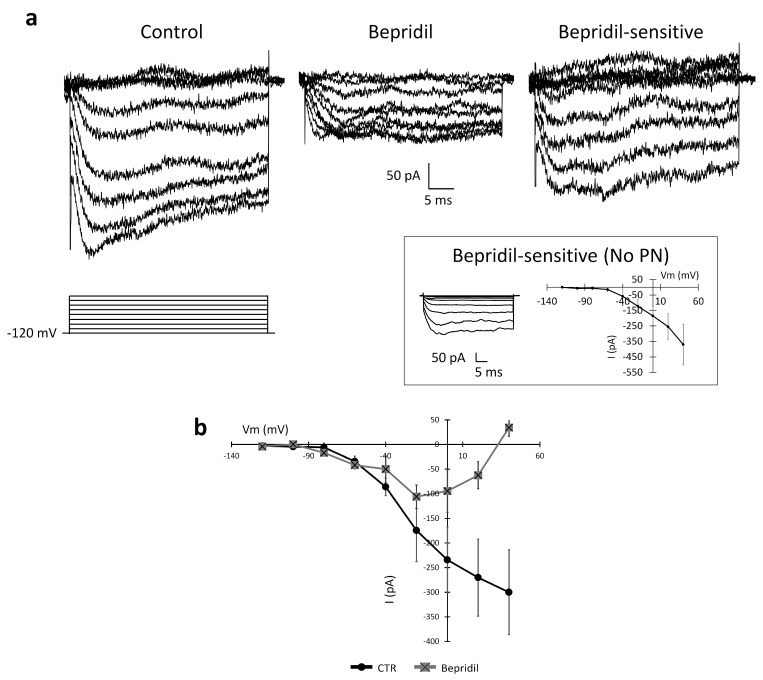
Inhibition of the inward current detected in the U251 edge cells by 100 µM bepridil, a competitive antagonist of the NCX. (**a**) Experimental traces elicited by depolarizing the voltage steps from −120 mV to +40 mV before (black traces) and after the local perfusion of bepridil (light grey traces). Inset: The PN leak subtraction method does not significantly alter the I–V plot of the inward current. (**b**) Mean I–V relationship before and after bepridil local perfusion (n = 6 cells).

**Figure 9 ijms-24-12673-f009:**
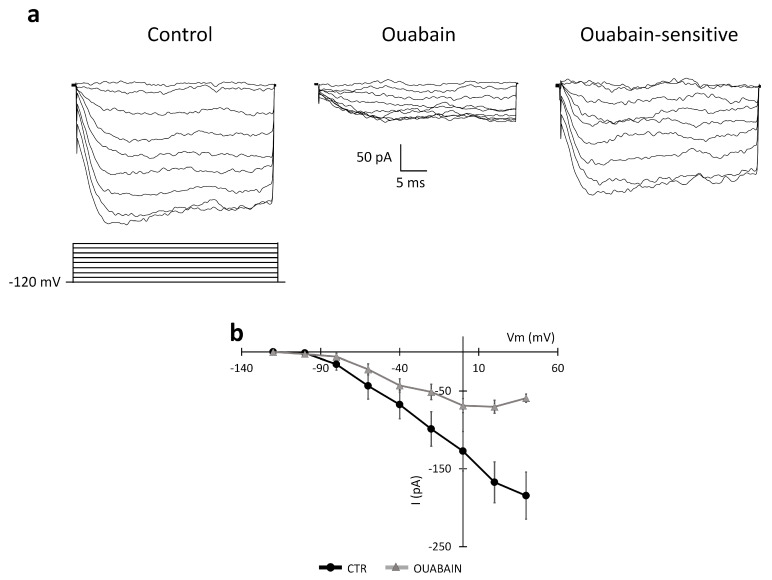
Effect of the Na-K ATPase blocker ouabain (perfused for about 10–15 min) on the NCX current in the U251 edge cells. (**a**) Experimental traces elicited by applying a voltage step depolarizing protocol from a holding potential of −120 mV (protocol shown at the bottom) in the control conditions (left traces) and after the local perfusion of ouabain 500 µM (central traces). The ouabain-sensitive current was obtained by digital subtraction (right traces). (**b**) Mean I–V relationships (n = 5 cells) recorded in the control conditions (CTR, black dots) and after the perfusion of ouabain (gray triangles).

**Figure 10 ijms-24-12673-f010:**
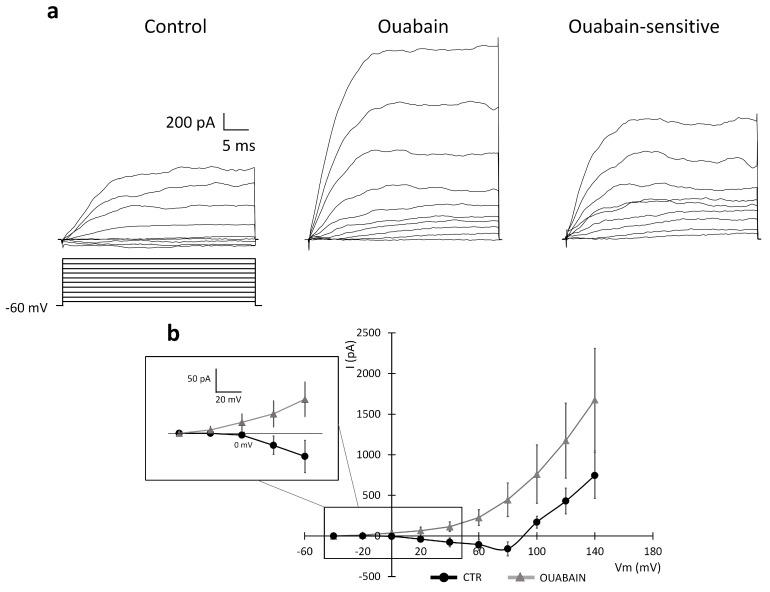
Effect of the Na-K ATPase blocker ouabain (perfused for about 10–15 min) on the BK current in U251 edge cells. (**a**) Experimental traces elicited by applying a voltage step depolarizing protocol from a holding potential of −60 mV (protocol shown at the bottom) in the control conditions (left traces) and after the local perfusion of ouabain 500 µM (central traces). The ouabain-sensitive current was obtained by digital subtraction (right traces). (**b**) Mean I–V relationships (n = 5 cells) recorded in the control conditions (CTR, dots) and the ouabain condition (ouabain, triangles). Insert shows the averaged I–V in the interval range between −40 mV and +40 mV at higher magnification to show the I–V shift after the local perfusion of ouabain.

**Figure 11 ijms-24-12673-f011:**
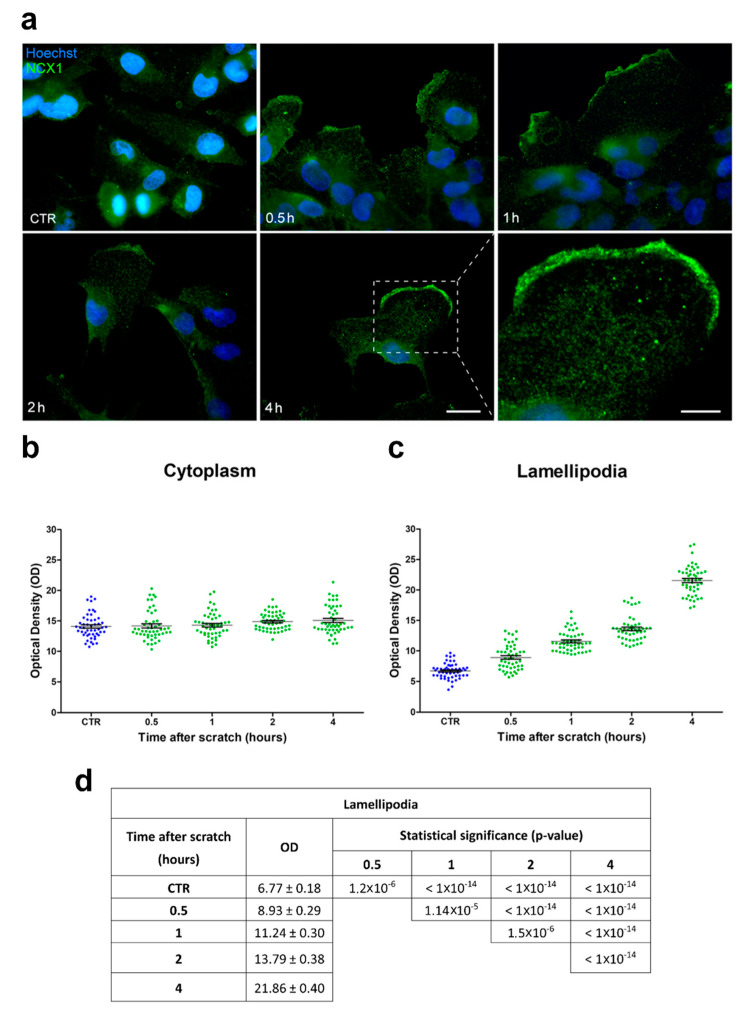
NCX1 expression in U251 cells was specifically upregulated in the lamellipodia after the scratch. (**a**) Representative micrographs showing NCX1 immunopositivity (green signal) in the cytoplasm and lamellipodia of cells located at the edge of the scratch in the control conditions (CTR) and 0.5, 1, 2, and 4 h after the scratch. Nuclear counterstaining with Hoechst 33258 (blue). Scale bar: 20 µm. Lower right panel: high-magnification (180×) micrograph showing the lamellipodia 4 h after the scratch. Scale bar: 10 µm. (**b**,**c**) Optical density (OD) scatter plots illustrating single-cell measurements acquired in the cytoplasm (**b**) and lamellipodia (**c**) in the control conditions (CTR) and at different time points after the scratch. Data are expressed as the mean ± SEM. (**d**) Lamellipodia mean OD values and relative statistical significance in the basal conditions (CTR) and 0.5, 1, 2, and 4 h after the scratch. Data are expressed as the mean ± SEM. *p*-values were calculated using one-way ANOVA, followed by the Bonferroni post hoc test.

**Figure 12 ijms-24-12673-f012:**
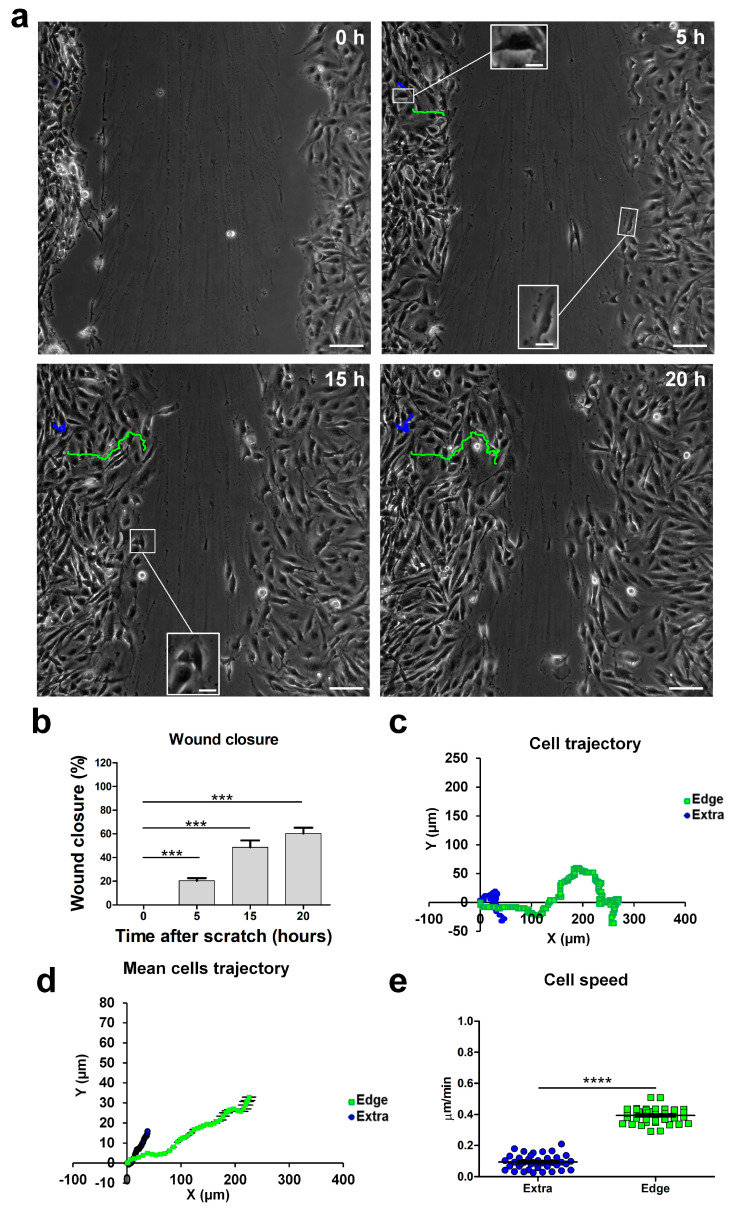
Blockade of NCX1 with nickel in U251 cells’ impaired migration trajectory and speed in the wound-healing assay. (**a**) Tracking of individual cells located in the extra-scratch (blue) and one at the edge of the wounded area (green) in the presence of nickel 200 µM. Images captured at 0, 5, 15, and 20 h after the wound scratch. Scale bar: 100 µm. Inset: a typical lamellipodia of a cell localized at the edge of the scratch; lamellipodia can be detected 5 h after the scratch, but they are less visible over time. Scale bar: 10 µm. (**b**) Wound closure percentage at 0, 5, 15, and 20 h after the wound scratch in the nickel condition. (**c**) Plotting of the trajectories of one individual cell located outside (extra, blue) and one individual cell at the edge of the scratch (edge, green). Images were analyzed with the Chemotaxis Tool ImageJ software plugin (Version 2.0). (**d**) Mean trajectories of the cells outside (extra, blue, n = 40 cells) and at the edge of the scratch (edge, green, n = 40 cells). (**e**) Scatter plot of the velocity (µm/min) of single cells outside (extra, blue, n = 40 cells) and at the edge of the scratch (edge, green, n = 40 cells). *p*-values: *** *p* < 0.001, and **** *p* < 0.0001.

**Figure 13 ijms-24-12673-f013:**
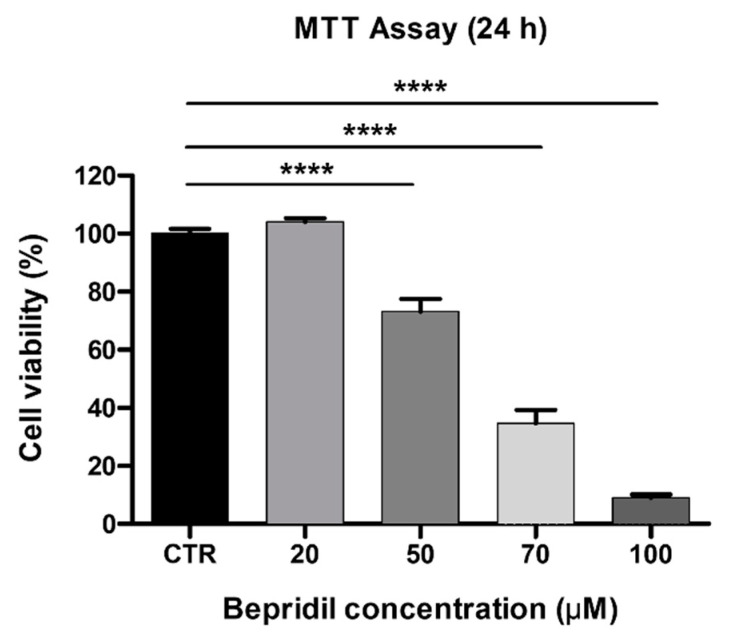
Effects of bepridil on cell viability/proliferation of U251 cells. Data obtained with the MTT vitality assay after 24 h exposure to increasing concentrations (20, 50, 70, and 100 μM) of bepridil. The relative cell viability is expressed as a percentage relative to the untreated control cells. Data are expressed as the mean ± SEM. *p*-values: **** *p* < 0.0001 compared to the control (100% cell viability).

**Figure 14 ijms-24-12673-f014:**
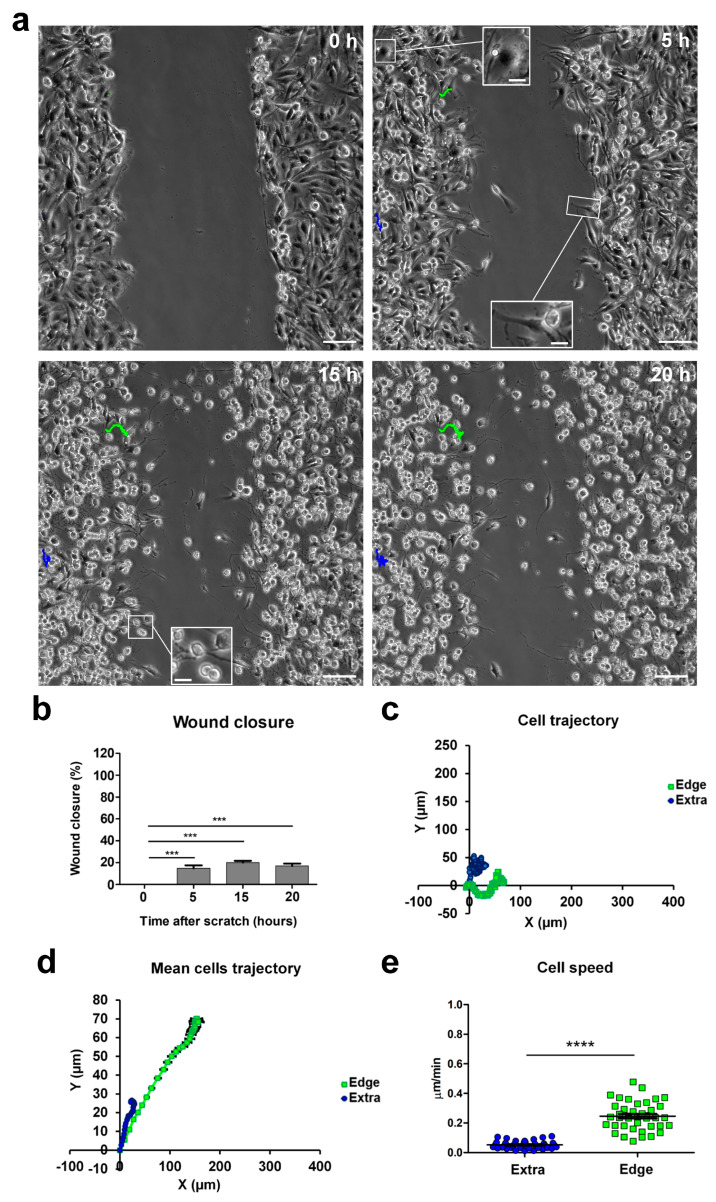
Blockade of NCX1 with bepridil in U251 cells’ impaired migration trajectory and speed in the wound-healing assay. (**a**) Tracking of individual cells located in the extra-scratch (blue) and at the edge of the wounded area (green) in the presence of bepridil 50 μM. Images captured at 0, 5, 15, and 20 h after the wound scratch. Scale bar: 100 µm. Inserts (scale bar: 10 µm) depicting lamellipodia as typical features of cells localized at the edge of the scratch; lamellipodia are evident 5 h after the scratch, but they are no longer detectable over time, i.e., 15 and 20 h after the scratch. (**b**) Wound closure percentage at 0, 5, 15, and 20 h after the wound scratch in the bepridil condition. (**c**) Plotting of the trajectories of one individual cell located outside (edge, blue) and one individual cell positioned at the edge of the scratch (edge, green). Images were analyzed by the Chemotaxis Tool ImageJ software plugin (Version 2.0). (**d**) Mean trajectories of cells outside (extra, blue, n = 40 cells) and at the edge of the scratch (edge, green, n = 40 cells). (**e**) Scatter plot of the velocity (µm/min) of single cells outside (extra, blue, n = 40 cells) and at the edge of the scratch (edge, green, n = 40 cells). *p*-values: *** *p* < 0.001, and **** *p* < 0.0001.

**Table 1 ijms-24-12673-t001:** Mean amplitude of the BK and KIR currents, measured at +140 mV and −120 mV, respectively, in cells recorded extra, intra, and at the edge of the wounded area. N.P. = not present.

Recording Position	BK (pA)	BK (pA/pF)	Kir (pA)	Kir (pA/pF)
Extra	113.84 ± 19.79 (n = 25)	11.59 ± 1.08 (n = 25)	N.P.	N.P.
Intra	547.41 ± 81.46 (n = 37)	19.36 ± 1.86 (n = 37)	−87.81 ± 12.41 (n = 37)	−4.62 ± 0.67 (n = 37)
Edge	820.64 ± 92.39 (n = 23)	27.20 ± 2.21 (n = 23)	−37.94 ± 6.71 (n = 19)	−0.88 ± 0.20 (n = 19)

## Data Availability

The datasets generated during and/or analyzed during the current study are not publicly available.

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
