# Peer review of "Role of Na+/Ca2+ Exchanger (NCX) in Glioblastoma Cell Migration (In Vitro)"

_ijms, 2023, doi:10.3390/ijms241612673_

Round 1

Reviewer 1 Report (New Reviewer)

The authors explore the role of Na+/Ca2+ exchanger in glioblastoma cell migration via using non-invasive perforated patch-clamp and live-imaging approaches on various GBM cell lines during a wound healing assay.The  date presented in the manuscript are suffiecient to support the conclusion. However, some minor flaws should be addressed before publication.

1. What does "NCX" mean in the title?

2. Please provide detailed information for the abbreviations when first appeared in the manuscript, such as BK.

Author Response

Reviewer 1

Comments and Suggestions for Authors

The authors explore the role of Na+/Ca2+ exchanger in glioblastoma cell migration via using non-invasive perforated patch-clamp and live-imaging approaches on various GBM cell lines during a wound healing assay.The  date presented in the manuscript are suffiecient to support the conclusion. However, some minor flaws should be addressed before publication.

  1. What does "NCX" mean in the title?

We thank the reviewer for addressing this point. Since we use “NCX” as the abbreviation for the Na+/Ca2+ exchanger, we put it in parenthesis.

  1. Please provide detailed information for the abbreviations when first appeared in the manuscript, such as BK.

We agree with the review. We have now carefully provided information for each abbreviation the first time they appear in the manuscript as suggested.

Reviewer 2 Report (New Reviewer)

Glioblastoma are among the most devastating tumors, also due to the high cell motility of glioblastoma cells. The activity of ion transporting proteins can contribute to cell migration. In this study, Brandalise et al. analyzed cation currents of glioblastoma cells in a 2D cell migration assay. Besides Kir and BK channels, they report the activity of NCX in migrating cells. They study the expression at the leading edge by immunostaining and the role of NCX by pharmacological inhibition.

Some controls are lacking for some of the conclusions and some changes are required to improve the manuscript.

1)    For most of the experiments: it is not clear what the difference between “edge” and “intra” is. Are “intra” not migrating anymore? Do they just have more empty space around while “edge” are still in contact with other cells?

2)    Figure 1d: Why do all cells move “upward” after 150 µM? If it were random, one would expect the mean to remain on the x-axis with only standard deviation increasing.

3)    Line 65: “When extracellular Ca2+ concentration rises..”. Do the authors mean intracellular? Otherwise it doesn’t seem to make sense, also in comparison with Na+ in the next sentences.

4)    Lines 155f: “confirming the undifferentiated depolarization state of the cell”: It is not clear what the authors want to convey with this – then all three “cell states” are undifferentiated? Are the “intra” even more undifferentiated then the “extra”?

5)    Line 161f: Could the larger cell surface be independent of being committed to migration, and just because there is more space in the open field of the scratch area?

6)    Fig. 3 and 4: Current traces should also be shown for “extra” to appreciated there are no (?) currents above background. As the cell surface is different between cells, the authors should also plot ion density.

7)    Line 187 and below: The term “overexpression” does not really fit.

8)    Lines 252 and 541: do the authors assume it is GBM-specific? What about any other cell line?

9)    Lines 350ff and Fig. 9: If the NCX now works “in reverse mode”, would not outward currents be expected?

10) Fig. 11: The immuno-staining lacks control of antibody specificity (even by secondary antibody alone). A knockout or at least siRNA control would be suited. At least co-staining with another plasma membrane protein that is not enriched in the leading edge.

11) NCX1 knockout or siRNA would also be good for to test the role of NCX in the migration assay apart from nickel (for bepridil, see below), is it feasible?

12) It is not clear why the authors use 50 µM bepridil if they find this concentration even kills 30% of the cells. Migration is certainly even more effected than survival and in Fig. 14a, hardly any cell looks healthy.

13) Lines 418ff: There is a significant effect of nickel on “extra” – even an increase in migration speed. But here NCX currents were not shown. So, what does this now mean?

14) The discussion is very close at the results. The authors could discuss more in comparison to other ion transporters or other cell lines (see below). Lines 548-550 is a word-by-word repetition of lines 507-509.

15) The proposed role of NCX in cell migration should be discussed, esp. regarding a study on the roles of NHE and VRAC in cell migration (Zhang et al., Nat Commun 2022).

16) As Cl transport is also required for volume regulation and there is no basal activity of Cl channels, the authors should mention the debated role of VRAC in gliobastoma migration (see Zhang as above, and Wong et al., J Cell Physiol; but in contrast Liu and Stauber, IJMS 2019, who also used a 2D cell migration assay with glioblastoma cells and found no effect).

17) The recent study by Michelucci et al., J Cell Physiol 2023, on Ca-activated K channels in glioblastoma cells should be mentioned.

Author Response

Reviewer 2

Comments and Suggestions for Authors

Glioblastoma are among the most devastating tumors, also due to the high cell motility of glioblastoma cells. The activity of ion transporting proteins can contribute to cell migration. In this study, Brandalise et al. analyzed cation currents of glioblastoma cells in a 2D cell migration assay. Besides Kir and BK channels, they report the activity of NCX in migrating cells. They study the expression at the leading edge by immunostaining and the role of NCX by pharmacological inhibition.

Some controls are lacking for some of the conclusions and some changes are required to improve the manuscript.

1)    For most of the experiments: it is not clear what the difference between “edge” and “intra” is. Are “intra” not migrating anymore? Do they just have more empty space around while “edge” are still in contact with other cells?

We fully agree with the reviewer. We have now better explained the criteria to discriminate between “edge” and “intra” cells in the manuscript by adding the information in the Results section (see lines 144-150) as follow: “GBM cells were classified as “edge cells” when in bright field they show a typical phenotype characterized by digitiform and lamellipodia-like structures polarized towards the wound healing, but they have not yet entered inside the scratch (see Fig. 1a). “Extra” cells are classified as cells that have not entered the scratch and do not have lamellipodia-like structure (they usually have a typical bipolar morphology). “Intra cells” are classified as the one who have entered already inside the wound healing.”

Based on our data, intra cells invade free space in the wound healing and search for contacts with other cells. When they do so they significantly decrease their migration rate. However, in this work we did not focus on the post-migratory part of the assay, but it will be subject of investigation for the upcoming follow-up.

2)    Figure 1d: Why do all cells move “upward” after 150 µM? If it were random, one would expect the mean to remain on the x-axis with only standard deviation increasing.

We thank the reviewer for noticing that. In the time lapse experiment, we have tracked cells on the bottom left corner of the field of view (FOV) going upward and the reason is practical: if we have tracked a cell from the bottom left corner and it was migrating downward, we would have lost it from the FOV after few hours. We consequently choose a starting point in the track along with an orientation of the migration of the cell that allowed us to follow it for the all recording time. We could have chosen the top right part of the FOV and track cells that were going downwards. To clarify this methodological approach we have added the following section in the method section (lines 698-701):

“Only cells that were present in the field of view (FOV) for the all recording time were included in the analysis of the time lapse for all the conditions applied. Usually, cells located on the bottom left corner of the FOV and with an upward direction in the migration were selected and tracked.”

3)    Line 65: “When extracellular Ca2+ concentration rises..”. Do the authors mean intracellular? Otherwise it doesn’t seem to make sense, also in comparison with Na+ in the next sentences.

We thank the reviewer for pointing-out this lack. We have accordingly modified the sentence (now in lines 68-69).

4)    Lines 155f: “confirming the undifferentiated depolarization state of the cell”: It is not clear what the authors want to convey with this – then all three “cell states” are undifferentiated? Are the “intra” even more undifferentiated then the “extra”?

Following the comment of the reviewer, we have deleted the last part of the sentence as the degree of differentiation was not fully investigated.

5)    Line 161f: Could the larger cell surface be independent of being committed to migration, and just because there is more space in the open field of the scratch area?

We thank the reviewer for the comment. Indeed, this is a possibility. However, at least for the recording period analyzed, cells that were not committed to migration maintain the same status and the same electrophysiological properties even when edge cells begin to migrate and more space becomes available. Following this point, we have toned-down the sentence as follow:

“Therefore, the high Cm and low Rm correlates with a larger cell surface and a bigger membrane conductance in cells committed to migration.”

6)    Fig. 3 and 4: Current traces should also be shown for “extra” to appreciated there are no (?) currents above background. As the cell surface is different between cells, the authors should also plot ion density.

We thank the reviewer for this point. Accordingly, in the figures 3 and 4, we added experimental traces, for example of “extra” condition (panels a), and the new scatter plots reporting the current density (panels e). Accordingly, we implemented the “results section” and Table 1.

7)    Line 187 and below: The term “overexpression” does not really fit.

Accordingly with the reviewer, we modify it.

8)    Lines 252 and 541: do the authors assume it is GBM-specific? What about any other cell line?

The reviewer is raising an interesting point here. In this work specifically, we have focused on the migration process of GBM cells as the role of this channel was not fully unveiled. It will be interesting to investigate for example its implication in the glia as there are reports suggesting a role of the NCX (in reverse mode) in migration of microglia (Noda et al.2013). We have added the following sentence in this regard in the discussion in line 551:

“Intriguingly it has also shown to be involved in the regulation of migration of microglia (Noda et al. 2013).”

Noda M, Ifuku M, Mori Y, Verkhratsky A. Calcium influx through reversed NCX controls migration of microglia. Adv Exp Med Biol. 2013;961:289-94. doi: 10.1007/978-1-4614-4756-6_24. PMID: 23224888.

9)    Lines 350ff and Fig. 9: If the NCX now works “in reverse mode”, would not outward currents be expected?

We thank the reviewer for underlining this point. Since ouabain was acutely perfused (10-15 minutes) and NCX current was reduced, we suggest a decrease in Ca2+ extrusion that positively activate BK and gBK current. We cannot exclude that, for longer perfusion time, a change in ion gradients could switch the exchanger to the reverse mode. For this reason, we better described the experiment indicating the perfusion time for ouabain and the corresponding sentence (line 399-401) was changed to:

“Therefore, these data suggest that a reduction in the forward mode of the NCX and the subsequent decrease in the Ca2+ extrusion, may indeed positively activate BK and gBK current. We cannot exclude that, for longer perfusion time, a change in ion gradients could switch the exchanger to the reverse mode.”

10) Fig. 11: The immuno-staining lacks control of antibody specificity (even by secondary antibody alone). A knockout or at least siRNA control would be suited. At least co-staining with another plasma membrane protein that is not enriched in the leading edge.

We thank the Reviewer for raising this issue. For clearness, it has to be highlighted that, as a good laboratory practice, we include an appropriate negative control at each experimental session, even though commercial, validated antibodies are usually employed. In detail, as negative control, some cell samples are incubated with PBS only, in the absence of the primary employed antibodies.

In particular, concerning our current manuscript, we performed immunofluorescent analyses including the negative control (cells incubated with PBS only, in absence of anti-NCX1 isoform primary antibody) and revealing a complete lack of immunoreactivity, therefore proving the antibody specificity (see below Figure a, depicting this latter experimental condition).

Moreover, following Reviewer suggestions, to further support our data, additionally proving the antibody specificity, we carried out novel, supplementary double-immunofluorescence experiments, investigating the expression and localization of NCX1 and β‐tubulin, being this latter a valuable microtubule marker. We chose to perform the double staining using this marker since the essential prerequisite for cancer cells to metastasize is the dramatic reorganization of their cytoskeleton (Ruggiero and Lalli, 2021). It had to be mentioned that literature data clearly established the central role played by actin polymerization and actin/myosin contractile forces in causing cell migration, a mechanism crucially involved in pathological processes of cancer cells migration and invasion-metastasis cascade. Less is known about the role of microtubules in this process and the available data are often controversial. In fact, large actin-based sheet-like protrusions called lamellipodia form the ‘front’ of the migrating cell. Actin polymerization at the leading edge creates protrusive force, which, together with adhesions and motor proteins drives the cell forward.

Furthermore, growing evidence suggests a close relationship between actin and microtubule organization with the microtubules possibly playing a significant role in protrusion formation and actin rearrangements (Etienne-Manneville, 2004; Ganguly et al., 2012; Woodham and Machesky, 2014). Notably, it has to be underline that in U251 cellular protrusions, an actin-rich region can be distinguished, which includes lamellipodia and filopodia, filled by a dense network of filamentous actin but barely containing any microtubules (Etienne-Manneville, 2004; Prahl et al., 2018).

In accordance with these above reported literature evidences, our supplementary data confirmed the time-dependent increase of NCX1 expression in the leading edge (as already stated in the submitted main manuscript) and revealed the β‐tubulin expression pattern, evidencing a well-structured cytoskeleton, providing the integrity of edge cells morphology. Notably, focusing on the leading border of edge cells, the absence of overlapping signals is manifest.

In the Figure below are depicted the most representative micrographs, showing (i) anti-NCX1 isoform primary antibody omission and (ii) the double immunofluorescence reaction for NCX1 (red signal) and β‐tubulin (green signal) at two crucial time points, i.e. 0.5 h and 4 h after the scratch, respectively.

See figure in attached file.

Figure. (a) Anti-NCX1 isoform primary antibody omission; DNA counterstaining with Hoechst 33258 - blue fluorescence. (b and c) Double immunocytochemical detection of NCX1 (red signal) and β‐tubulin (green signal) by fluorescence microscopy 0.5 h and 4 h after the scratch, respectively. DNA counterstaining was performed using Hoechst 33258 (blue fluorescence).

Etienne-Manneville S. Actin and microtubules in cell motility: which one is in control? Traffic. 2004 Jul;5(7):470-7. doi: 10.1111/j.1600-0854.2004.00196.x. PMID: 15180824.

Ganguly A, Yang H, Sharma R, Patel KD, Cabral F. The role of microtubules and their dynamics in cell migration. J Biol Chem. 2012 Dec 21;287(52):43359-69. doi: 10.1074/jbc.M112.423905.

Woodham EF, Machesky LM. Polarised cell migration: intrinsic and extrinsic drivers. Curr Opin Cell Biol. 2014 Oct; 30:25-32. doi: 10.1016/j.ceb.2014.05.006.

Prahl LS, Bangasser PF, Stopfer LE, Hemmat M, White FM, Rosenfeld SS, Odde DJ. Microtubule-Based Control of Motor-Clutch System Mechanics in Glioma Cell Migration. Cell Rep. 2018 Nov 27;25(9):2591-2604.e8. doi: 10.1016/j.celrep.2018.10.101.

Ruggiero, C., Lalli, E. Targeting the cytoskeleton against metastatic dissemination. Cancer Metastasis Rev 40, 89–140 (2021). https://doi.org/10.1007/s10555-020-09936-0

11) NCX1 knockout or siRNA would also be good for to test the role of NCX in the migration assay apart from nickel (for bepridil, see below), is it feasible?

We thank the reviewer for the suggestion. Unfortunately, we do not have in our lab the facility to produce siRNA or the KO for the NCX1. We are however, currently collaborating with a lab to produce a conditional KO for the NCX1 to transiently silencing and reactivating the exchanger in our follow-up paper.

12) It is not clear why the authors use 50 µM bepridil if they find this concentration even kills 30% of the cells. Migration is certainly even more affected than survival and in Fig. 14a, hardly any cell looks healthy.

We thank the reviewer for raising this issue. It has to be mentioned that several investigations proved that bepridil is able to induce cytotoxicity in different human tumor cell lines in a dose-dependent manner (Lee et al., 1995; Liu et al. 2022). Furthermore, recent data revealed the dose-dependent cytotoxicity of bepridil on both U87 and U251 cells leading to significant cell reduction. Specifically, U87 and U251 cell viability was reduced to 30-40%, 6-7%, and 3-4% of control level by 20‐, 50‐, and 100 μM bepridil. In particular, the concentrations of 18.6±1.1 and 14.9±1.1 µM were demonstrated to be a half maximal inhibitory concentration, able to induce cell death, causing a significant decrease (about 50%) in the number of U87 and U251 living proliferating cells, respectively (Hu et al., 2019). Nonetheless, it has to be underlined that all these findings, showing bepridil toxicity, referred to prolonged time of exposure, i.e. 48 and 72 hours.

Concerning our data, it has to be taken into careful consideration that the exposure time to bepridil lasted 24 hours, therefore the treatment duration was shorter than those reported in previous studies. Additionally, it has to be highlighted that the bepridil dose employed in our current study was selected based on MTT vitality assay results. In particular, our findings obtained using U251 cells after 24 hours exposure to increasing concentrations of bepridil (20, 50, 70 and 100 μM), revealed that, interestingly, the tested dose of 20 μM unaffected cell viability, which persisted about 100%. Indicating a dose dependent cytotoxicity, the higher concentration of 50 μM was able to cause 30% cell viability decrease, while the concentration required to produce half maximal inhibition (IC50) of cell viability was about 60 μM. Therefore, having clear in mind the goal to clarify the role of NCX in U251 cell migration/invasiveness, we decided to chose the subtoxic concentration of 50 μM, leading to a mild reduction in the number of living/proliferation cells, as the proper suitable concentration to be used in the following experiments.

Indeed, in the text, we explained the choice of the employed bepridil concentration (see Results section).

Lee YS, Sayeed MM, Wurster RD. Intracellular Ca2+ mediates the cytotoxicity induced by bepridil and benzamil in human brain tumor cells. Cancer Lett. 1995 6;88:87-91. doi: 10.1016/0304-3835(94)03619-t.

Liu, Z.; Cheng, Q.; Ma, X.; Song, M. Suppressing Effect of Na+/Ca2+ Exchanger (NCX) Inhibitors on the Growth of Melanoma Cells. Int. J. Mol. Sci. 2022, 23, 901. https://doi.org/10.3390/ ijms23020901

Hu HJ, Wang SS, Wang YX, Liu Y, Feng XM, Shen Y, Zhu L, Chen HZ, Song M. Blockade of the forward Na+/Ca2+ exchanger suppresses the growth of glioblastoma cells through Ca2+-mediated cell death. Br J Pharmacol. 2019 Aug;176(15):2691-2707. doi: 10.1111/bph.14692.

13) Lines 418ff: There is a significant effect of nickel on “extra” – even an increase in migration speed. But here NCX currents were not shown. So, what does this now mean?

We thank the reviewer for the constructive comment. We have made a typo mistake saying the migration speed was higher in nickel condition vs control but the data are showing the opposite:

"The migration speed of cells located both outside the scratch and at the edge in the nickel condition was slower compared to the control condition. In particular, for “extra” cells the migration speed was 50 % higher than in control conditions (extra: 0.09 ± 0.01 µm/min in nickel vs extra control 0.18 ± 0.01 µm/min, n = 40 for both experimental conditions, p value = 1.14 * 10-09, unpaired t-test, Figure 12d)."

We have now substituted “higher” with “lower” (line 459).

Now, we can speculate that the effect of Nickel on the migration speed in “extra” cells can be due to the fact it acts as a blocker for some of calcium channels (see Zamponi et al. 1996) further reducing the intracellular amount of calcium but we do not have data ourself on this regards.

Zamponi, G. W., E. Bourinet, and T. P. Snutch. "Nickel block of a family of neuronal calcium channels: subtype-and subunit-dependent action at multiple sites." The Journal of membrane biology 151 (1996): 77-90.

14) The discussion is very close at the results. The authors could discuss more in comparison to other ion transporters or other cell lines (see below). Lines 548-550 is a word-by-word repetition of lines 507-509.

We thank the reviewer for the constructive criticism on the discussion session. We have rewritten the second sentence to avoid word-by-word repetition. The discussion has been rebuilt integrating all the literature indicated below and we have reduced the part linked to our results as suggested.

15) The proposed role of NCX in cell migration should be discussed, esp. regarding a study on the roles of NHE and VRAC in cell migration (Zhang et al., Nat Commun 2022).

See point 16.

16) As Cl transport is also required for volume regulation and there is no basal activity of Cl channels, the authors should mention the debated role of VRAC in gliobastoma migration (see Zhang as above, and Wong et al., J Cell Physiol; but in contrast Liu and Stauber, IJMS 2019, who also used a 2D cell migration assay with glioblastoma cells and found no effect).

We thank the reviewer for suggesting this group of papers on the topic of the chloride dynamics. We have added all of them in a dedicated sentence in the discussion section (lines 531-536):

“More recently, other conductances have been described to play a major role in the  regulation of the chloride dynamics such as the volume-regulated anion channel (VRAC/LRRC8) (Zhang et al. 2022, Wong et al. 2017). However, the importance of the last channel in GBM migration is still under debate (Liu & Stauber, 2019) due to possible variabilities in both the cell line adopted between these works and the migration model used.” 

Liu, Tianbao, and Tobias Stauber. "The volume-regulated anion channel LRRC8/VRAC is dispensable for cell proliferation and migration." International Journal of Molecular Sciences 20, no. 11 (2019): 2663.

Wong, Raymond, Wenliang Chen, Xiao Zhong, James T. Rutka, Zhong‐Ping Feng, and Hong‐Shuo Sun. "Swelling‐induced chloride current in glioblastoma proliferation, migration, and invasion." Journal of cellular physiology 233, no. 1 (2018): 363-370.

Zhang, Yuqi, Yizeng Li, Keyata N. Thompson, Konstantin Stoletov, Qinling Yuan, Kaustav Bera, Se Jong Lee et al. "Polarized NHE1 and SWELL1 regulate migration direction, efficiency and metastasis." Nature communications 13, no. 1 (2022): 6128.

17) The recent study by Michelucci et al., J Cell Physiol 2023, on Ca-activated K channels in glioblastoma cells should be mentioned.

We truly appreciate the suggestion of the reviewer. Indeed, this is a fundamental manuscript. We have consequently mentioned the above paper (lines 618-620) as follow:

“GBM cells can adopt other strategies to increase the cytosolic calcium during migration such as the opening of mechanosensitive potassium channels through hypotonicity as reported in Michelucci et al. 2023 (see also Caramia et al. 2019).”

Caramia, Martino, Luigi Sforna, Fabio Franciolini, and Luigi Catacuzzeno. "The volume-regulated anion channel in glioblastoma." Cancers 11, no. 3 (2019): 307.

Michelucci, Antonio, Luigi Sforna, Angela Di Battista, Fabio Franciolini, and Luigi Catacuzzeno. "Ca2+‐activated K+ channels regulate cell volume in human glioblastoma cells." Journal of Cellular Physiology.

Reviewer 3 Report (New Reviewer)

The manuscript entitled “Role of Na+/Ca2+ exchanger NCX in glioblastoma cell migration 2 (in vitro)” by Federico Brandalise and coworkers is an interesting manuscript devoted to the study of the ionic dynamics underlying the migration of human glioblastoma cells through a gap induced in a semiconfluent 2D culture (scratch assay). According to their results glioblastoma cell migration across the gap require NCX accumulation and activity at the leading edge of the migrating cell. Interestingly, they found that blocking the forward Na+/Ca2+ exchanger by pharmacological manipulation significantly slowed glioblastoma cell migration across the gap. The findings described in the present manuscript nicely complement the results of previous experiments by Hu et al. showing that inhibiting the forward Na+/Ca2+ exchanger induces glioblastoma cell death both in vitro and in vivo.

When NCX is described in the introduction, the authors do not mention the published articles by Song et al. (doi: 10.1111/bph.12691 and Lu et al. (doi: 10.1111/bph.14692). Both articles are closely related to the experiments described in the present manuscript and both should be mentioned in the introduction and the discussion. In the current manuscript, only Lu et al. is mentioned in the discussion to support the statement that "blockade of the forward 557 Na+/Ca2+ exchanger by nickel and bepridil significantly slowed the time for wound closure". However, Lu et al. did not report data on glioblastoma cell migration.

The major limitation I see in the experiments described is that, for good experimental reasons, they were limited to 2D cultures and the scratch assays results have a relatively poor predictive value of glioblastoma cells behavior in vivo. This aspect should be thoroughly considered in the discussion.

Have the authors tested if NCX2 and NCX3 are expressed by any of the cell lines or by the primary cell cultures they employed? If they are not expressed why the authors do not clearly indicate NCX1 in place of NCX thorough the manuscript.?

Hu et al. (https://doi.org/10.1111/bph.14692) found that viability of U251 cells was reduced to 30–40%, 6–7%, and 3–4% of control level by 20-, 50-, and 100-μM bepridil, while in your figure 13 the cytotoxic effect on the same cell line was significant only at concentrations of bepridil > 70 microM, please explain.

Author Response

Reviewer 3

Comments and Suggestions for Authors

 The manuscript entitled “Role of Na+/Ca2+ exchanger NCX in glioblastoma cell migration 2 (in vitro)” by Federico Brandalise and coworkers is an interesting manuscript devoted to the study of the ionic dynamics underlying the migration of human glioblastoma cells through a gap induced in a semiconfluent 2D culture (scratch assay). According to their results glioblastoma cell migration across the gap require NCX accumulation and activity at the leading edge of the migrating cell. Interestingly, they found that blocking the forward Na+/Ca2+ exchanger by pharmacological manipulation significantly slowed glioblastoma cell migration across the gap. The findings described in the present manuscript nicely complement the results of previous experiments by Hu et al. showing that inhibiting the forward Na+/Ca2+ exchanger induces glioblastoma cell death both in vitro and in vivo.

When NCX is described in the introduction, the authors do not mention the published articles by Song et al. (doi: 10.1111/bph.12691 and Lu et al. (doi: 10.1111/bph.14692). Both articles are closely related to the experiments described in the present manuscript and both should be mentioned in the introduction and the discussion. In the current manuscript, only Lu et al. is mentioned in the discussion to support the statement that "blockade of the forward 557 Na+/Ca2+ exchanger by nickel and bepridil significantly slowed the time for wound closure". However, Lu et al. did not report data on glioblastoma cell migration.

We thank the reviewer for suggesting us these fundamental publications. Indeed, they are closely related to our work and it is important to mention them in the paper. We have added them in the introduction (lines 62-64) as suggested with the following sentence: “NCX has been already subject of investigation in GBM, related to the activity of the TRPC channel (Song et al. 2014) or to Ca2+-mediated cell death (Hu et al. 2019). However, there is a paucity of reports on its role on GBM cell migration.” 

Hu HJ, Wang SS, Wang YX, Liu Y, Feng XM, Shen Y, Zhu L, Chen HZ, Song M. Blockade of the forward Na+ /Ca2+ exchanger suppresses the growth of glioblastoma cells through Ca2+ -mediated cell death. Br J Pharmacol. 2019 Aug;176(15):2691-2707. doi: 10.1111/bph.14692. Epub 2019 Jun 17. PMID: 31034096; PMCID: PMC6609550.

Song M, Chen D, Yu SP. The TRPC channel blocker SKF 96365 inhibits glioblastoma cell growth by enhancing reverse mode of the Na(+) /Ca(2+) exchanger and increasing intracellular Ca(2+). Br J Pharmacol. 2014 Jul;171(14):3432-47. doi: 10.1111/bph.12691. PMID: 24641279; PMCID: PMC4105931.

The major limitation I see in the experiments described is that, for good experimental reasons, they were limited to 2D cultures and the scratch assays results have a relatively poor predictive value of glioblastoma cells behavior in vivo. This aspect should be thoroughly considered in the discussion.

We thank the reviewer for the important suggestion: in agreement, the following sentence has been integrated in the discussion (lines 600-604): “To further confirm the role of NCX in glioblastoma, in vivo experiments should also be considered as in 2D cultures and, by using the scratch assay, we might have relatively limited predictive value of glioblastoma cells behavior in a more complex and physiological environment.”

Have the authors tested if NCX2 and NCX3 are expressed by any of the cell lines or by the primary cell cultures they employed? If they are not expressed why the authors do not clearly indicate NCX1 in place of NCX through the manuscript.?

We thank the reviewer for pointing out the topic. We did not test for the other subunit isoforms. One of the reasons resides in the fact that the mRNA expression of SLC8A2 and SLC8A3 are significantly lower than SLC8A1 in GBM compared to other grades of glioma (see dataset extracted from “GLIOVIS”). Additionally, SLC8A1 is the only isoform to have an mRNA expression comparable to non-tumor tissue while all the others have a significantly lower expression in GBM versus non-tumor tissue.  Functionally speaking however, it is known that the blockers we used (nickel, bepridil) are blockers of all the isoforms, so the conclusion on the role of NCX1 can be potentially extended to NCX2 and NCX3 as well.  

See the file attached for images:

  • mRNA expression of SLC8A3 taken from GloVis ("GlioVis data portal for visualization and analysis of brain tumor expression datasets" (Bowman R. et al., Neuro-Oncology 2017). Each data point is a bulk mRNA analysis from a single patient.
  • mRNA expression of SCL8A2 taken from GlioVis  ("GlioVis data portal for visualization and analysis of brain tumor expression datasets" (Bowman R. et al., Neuro-Oncology 2017).Each data point is a bulk mRNA analysis from a single patient.
  • mRNA expression of SCL8A1 taken from GlioVis  ("GlioVis data portal for visualization and analysis of brain tumor expression datasets" (Bowman R. et al., Neuro-Oncology 2017). Each data point is a bulk mRNA analysis from a single patient.

Note how the mRNA level of SCL8A1 is not downregulated in GBM as compared to the non-tumor tissue as well as other brain tumors.

Hu et al. (https://doi.org/10.1111/bph.14692) found that viability of U251 cells was reduced to 30–40%, 6–7%, and 3–4% of control level by 20-, 50-, and 100-μM bepridil, while in your figure 13 the cytotoxic effect on the same cell line was significant only at concentrations of bepridil > 70 microM, please explain.

We thank the reviewer for raising this issue. It has to be mentioned that several investigations proved that bepridil is able to induce cytotoxicity in different human tumor cell lines in a dose-dependent manner (Lee et al., 1995; Liu et al. 2022). Recent data revealed the dose-dependent cytotoxicity of bepridil on both U87 and U251 cells leading to significant cell reduction. Specifically, U87 and U251 cell viability was reduced to 30-40%, 6-7%, and 3-4% of control level by 20‐, 50‐, and 100 μM bepridil. In particular, the concentrations of 18.6±1.1 and 14.9±1.1 µM were demonstrated to be a half maximal inhibitory concentration, able to induce cell death, causing a significant decrease (about 50%) in the number of U87 and U251 living proliferating cells, respectively (Hu et al., 2019).

Nonetheless, it has to be remarkably emphasized that all the above reported data displayed bepridil toxicity assessed after prolonged time of exposure, i.e. 48 and 72 hours. Differently, our current results were obtained after a shorter exposure duration, i.e. 24 hours, revealing the effects of bepridil on cell viability/proliferation of U251 cells. Therefore, it is reasonable to hypothesize that bepridil induced cytotoxicity not only in a dose dependent manner (as  broadly demonstrated in literature ) but also in a time dependent manner, with the dose of 70 microM able to cause a significant viability reduction (about 65%), while the lowest tested doses (25 and 50 microM) remaining ineffective or slightly toxic in the short-term exposure experiment. Reasoning about these dose-dependent effect determined after 24 hours-exposure by MTT assay, we then selected the concentration of 50 μM Bepridil, leading to a mild reduction (about 30%) in the number of living/proliferation cells, for the subsequent time-lapse microscopy analysis.

Lee YS, Sayeed MM, Wurster RD. Intracellular Ca2+ mediates the cytotoxicity induced by bepridil and benzamil in human brain tumor cells. Cancer Lett. 1995 6;88:87-91. doi: 10.1016/0304-3835(94)03619-t.

Liu, Z.; Cheng, Q.; Ma, X.; Song, M. Suppressing Effect of Na+/Ca2+ Exchanger (NCX) Inhibitors on the Growth of Melanoma Cells. Int. J. Mol. Sci. 2022, 23, 901. https://doi.org/10.3390/ ijms23020901

Hu HJ, Wang SS, Wang YX, Liu Y, Feng XM, Shen Y, Zhu L, Chen HZ, Song M. Blockade of the forward Na+/Ca2+ exchanger suppresses the growth of glioblastoma cells through Ca2+-mediated cell death. Br J Pharmacol. 2019 Aug;176(15):2691-2707. doi: 10.1111/bph.14692.

Round 2

Reviewer 2 Report (New Reviewer)

The authors have sufficiently addressed most concerns and the revision has clearly strengthened the manuscript.

Just minor points remain, that can be addressed textually:

- Previous point 10: absence of staining without primary antibody does not prove specificity of the primary antibody at all, but of the secondary. Specificity of primary antibody can only be shown with lack of specific target (i.e. knock-out or efficient knock-down). Many “commercial, validated antibodies” yield the same signal in KO control. So, if the authors know of any study that proves the specificity of the NCX1 antibody, they should cite it. (It is unclear what the argument with the costaining of tubulin means, of course the NCX1 (?) staining of the leading edge increases with time.)

- Previous point 11: it is rather surprising that the authors cannot obtain siRNA or do a CRISPR/Cas KO in their cell lines, but then such experiments could be mentioned as an outlook in the Discussion.

- Previous point 12: it was understood what was written in the manuscript. Yet, the 50 µM is clearly not a “subtoxic concentration” as it kills 20% of the cells within 24 h (Fig. 13) and we see what the cells look like in Fig. 14 (in comparison to Fig. 12). It is not required that the authors revise, the reader will not be surprised that migration is affected in these cells.

- Lines 531-536: “However, the importance of the last channel…” meaning VRAC/LRRC8. Rather name the channel or “latter”.

- There is a shift in the reference numbers (at least for most new references):There is a shift in the reference numbers (at least for most new references): Caramia et al is ref 27 in text, but ref 28 in bibliography; Zhang et al is (28/29); Wong et al (29/30); Liu et al (30/31); Noda et al (32/33).

Author Response

Comments and Suggestions for Authors

The authors have sufficiently addressed most concerns and the revision has clearly strengthened the manuscript.

Just minor points remain, that can be addressed textually:

- Previous point 10: absence of staining without primary antibody does not prove specificity of the primary antibody at all, but of the secondary. Specificity of primary antibody can only be shown with lack of specific target (i.e. knock-out or efficient knock-down). Many “commercial, validated antibodies” yield the same signal in KO control. So, if the authors know of any study that proves the specificity of the NCX1 antibody, they should cite it. (It is unclear what the argument with the costaining of tubulin means, of course the NCX1 (?) staining of the leading edge increases with time.)

We fully agree with the reviewer on this point. Since we did not test the antibody on a KO model for NCX1, we cannot prove the specificity of it. For this reason, we have removed the term “specific” from the chapter title 2.3. The CRISPR/Cas9 KO model will be handy in our follow up to make the proper control suggested by the reviewer. Meanwhile, as suggested by the reviewer, we report here some studies where the specificity of the mentioned antibody was confirmed through the use of negative control without primary and/or secondary antibody and blocking the nonspecific cross-reactivity by incubating the samples with BSA:

Wan H, Gao N, Lu W, Lu C, Chen J, Wang Y, Dong H. NCX1 coupled with TRPC1 to promote gastric cancer via Ca2+/AKT/β-catenin pathway. Oncogene. 2022 Aug;41(35):4169-4182. doi: 10.1038/s41388-022-02412-9. Epub 2022 Jul 26. PMID: 35882979; PMCID: PMC9418000.

Stern MD, Maltseva LA, Juhaszova M, Sollott SJ, Lakatta EG, Maltsev VA. Hierarchical clustering of ryanodine receptors enables emergence of a calcium clock in sinoatrial node cells. J Gen Physiol. 2014 May;143(5):577-604. doi: 10.1085/jgp.201311123. PMID: 24778430; PMCID: PMC4003189.

Dan P, Lin E, Huang J, Biln P, Tibbits GF. Three-dimensional distribution of cardiac Na+-Ca2+ exchanger and ryanodine receptor during development. Biophys J. 2007 Oct 1;93(7):2504-18. doi: 10.1529/biophysj.107.104943. Epub 2007 Jun 8. PMID: 17557789; PMCID: PMC1965441.

To clarify this point we added in Materials and Methods section the following sentence:

“The specificity of the antibody has been extensively described in different models (Wan et al., 2022; Stern et al., 2014; Dan et al., 2007)”.

Furthermore, Frank et al., 1992 described that the antibody screening was carried out by ELISA assays.

Frank JS, Mottino G, Reid D, Molday RS, Philipson KD. Distribution of the Na(+)-Ca2+ exchange protein in mammalian cardiac myocytes: an immunofluorescence and immunocolloidal gold-labeling study. J Cell Biol. 1992 Apr;117(2):337-45. doi: 10.1083/jcb.117.2.337. PMID: 1373142; PMCID: PMC2289429.

- Previous point 11: it is rather surprising that the authors cannot obtain siRNA or do a CRISPR/Cas KO in their cell lines, but then such experiments could be mentioned as an outlook in the Discussion.

We thank the reviewer for stressing the importance of the NCX silencing at the mRNA level. We fully agree with this point and we are going to include this approach along with the pharmacological one in the upcoming follow up work. Followed the reviewer suggestion we have mentioned it as an outlook in the discussion as follow:

“Since NCX is inhibited by external perfusion of nickel, we managed to isolate the NCX-mediated component (i.e., the nickel-sensitive component). To avoid potential incomplete blocks of the NCX, future experiments will also replicate our findings using a CRISPR/Cas9 KO model for the same cell lines”.

- Previous point 12: it was understood what was written in the manuscript. Yet, the 50 µM is clearly not a “subtoxic concentration” as it kills 20% of the cells within 24 h (Fig. 13) and we see what the cells look like in Fig. 14 (in comparison to Fig. 12). It is not required that the authors revise, the reader will not be surprised that migration is affected in these cells.

We thank the reviewer for clarifying this point. We agree that the assessment of cells status along with the experiment results can be evaluated by the reader.

- Lines 531-536: “However, the importance of the last channel…” meaning VRAC/LRRC8. Rather name the channel or “latter”.

We thank the reviewer for this suggestion. We modified the sentence as follow: “However, the importance of the VRAC/LRRC8 channel in GBM migration is still under debate [31] due to possible variabilities in both the cell line adopted between these works and the migration model used.”

- There is a shift in the reference numbers (at least for most new references):There is a shift in the reference numbers (at least for most new references): Caramia et al is ref 27 in text, but ref 28 in bibliography; Zhang et al is (28/29); Wong et al (29/30); Liu et al (30/31); Noda et al (32/33).

We thank the reviewer for underling this mistake. We corrected the mistake accordingly. 

This manuscript is a resubmission of an earlier submission. The following is a list of the peer review reports and author responses from that submission.

Round 1

Reviewer 1 Report

The Ms entitled “Invasion in glioblastoma: committed or not committed cells, 2 that is the question! The key role of the Na+/Ca2+ exchanger 3 NCX 4”” by Brandalise et al addresses the expression of the Na/Ca transporter NCX in cultured glioblastoma cell lines inside, outside and at the edge of the wounded area, investigates its electrophysiological properties, and its role in cell migration. The authors found several differences in the properties of these three groups of cells, including the membrane capacitance, the membrane resistance and the resting membrane potential. In edge cells the authors selectively record an inward current that increases with membrane depolarization and has an activation time constant of few tens of milliseconds. Based on the effects of bepridil and ouabain, it is suggested that this current is carried by the NCX transporter working in the forward mode. Finally, based on the effects of bepridil on cell migration it is concluded that the NCX transporter is important for the invasion strategy of these cells.

The study reaches interesting results such as the selective expression of NCX transporter in migrating cells only, and the role of this transporter in the migration process. However, there are some major limitations with regard to the electrophysiologic characterization of the NCX transportes that makes one doubt on the reliability of the results. First, it is not at all clear why the current carried by this transporter should increase with depolarization up to values as high as 140 mV. High depolarizations are expected to reduce or even revert the driving force for this transporter, since at each cycle a positive net charge is moved inside the cell, and this process is expected to have a voltage dependence opposite to that shown by the authors. Looking at the literature, the authors can easily find that the I-V relationship produced by this transporter is very different from the one they found (see for example Hobai, Pflugers Arch. 433: 455–463 1997). Second, standard methods to isolate the NCX current include the use of its blocker nickel, or the removal of extracellular Na ions to inhibit it, but the authors do not test these properties. Given the very peculiar and unusual properties of the current, I suspect that it may originate from an overcompensation of the cell capacitive peak. In fact, the capacitance and membrane resistance properties of GBM cells (C=30 pF, Rm=1 GOhm) would result in a relaxation of the capacitive peak of (Rm*C=) 30 ms, very similar to the relaxation properties of the current recorded.

Since the functional expression and electrophysiologic characterization of the NCX transporter is a central point of the Ms, I think it cannot be considered for publication until these doubts are fully dispelled.

Minor points

1.    For the level of the Ms, the title should be more simple and less existentialist. Something like ‘Role of Na+/Ca2+ exchanger NCX in glioblastoma cell migration (in vitro)’ would look more appropriate.

2.      P. 318-320. “After ouabain, the I/V relationship was shifted to the right of about 20 mV (insert in Figure 8)”. It is difficult to get how this figure (20 mV) has come out by looking at the inset.  

3.    Conclusion. Conclusion is too wordy and way too much speculative. The weakness of many of the results allows neither of them. My suggestion is to halve the discussion and refrain from any urge of speculation.

The English need to be improved. 

Author Response

Comments and Suggestions for Authors

The Ms entitled “Invasion in glioblastoma: committed or not committed cells, 2 that is the question! The key role of the Na+/Ca2+ exchanger 3 NCX 4”” by Brandalise et al addresses the expression of the Na/Ca transporter NCX in cultured glioblastoma cell lines inside, outside and at the edge of the wounded area, investigates its electrophysiological properties, and its role in cell migration. The authors found several differences in the properties of these three groups of cells, including the membrane capacitance, the membrane resistance and the resting membrane potential. In edge cells the authors selectively record an inward current that increases with membrane depolarization and has an activation time constant of few tens of milliseconds. Based on the effects of bepridil and ouabain, it is suggested that this current is carried by the NCX transporter working in the forward mode. Finally, based on the effects of bepridil on cell migration it is concluded that the NCX transporter is important for the invasion strategy of these cells.

The study reaches interesting results such as the selective expression of NCX transporter in migrating cells only, and the role of this transporter in the migration process. However, there are some major limitations with regard to the electrophysiologic characterization of the NCX transportes that makes one doubt on the reliability of the results. First, it is not at all clear why the current carried by this transporter should increase with depolarization up to values as high as 140 mV. High depolarizations are expected to reduce or even revert the driving force for this transporter, since at each cycle a positive net charge is moved inside the cell, and this process is expected to have a voltage dependence opposite to that shown by the authors. Looking at the literature, the authors can easily find that the I-V relationship produced by this transporter is very different from the one they found (see for example Hobai, Pflugers Arch. 433: 455–463 1997). Second, standard methods to isolate the NCX current include the use of its blocker nickel, or the removal of extracellular Na ions to inhibit it, but the authors do not test these properties. Given the very peculiar and unusual properties of the current, I suspect that it may originate from an overcompensation of the cell capacitive peak. In fact, the capacitance and membrane resistance properties of GBM cells (C=30 pF, Rm=1 GOhm) would result in a relaxation of the capacitive peak of (Rm*C=) 30 ms, very similar to the relaxation properties of the current recorded.

Since the functional expression and electrophysiologic characterization of the NCX transporter is a central point of the Ms, I think it cannot be considered for publication until these doubts are fully dispelled.

We agree with the reviewer that the current/voltage relationship of the inward current shown by the edge cells is atypical. However, we are sure that this current is not an artifact for the following reasons:

  1. We isolated the NCX current in the same experimental condition in three different laboratories (Lab of Neurobiology and Integrated Physiology, University of Pavia; Lab of Biophysics and Ion Channel Physiology, University of Pavia; Lab of Cellular and Molecular Physiology, University of Milan) recording the current with the same biophysical properties described in the paper (data not shown); 
  2. The NCX current is present at the edge on different cell lines (U251, mesenchymal GSCs and U87 cells – Fig. 5) and with the same biophysical properties; 
  3. The membrane capacitance did not change in cells recorded at the edge and intra scratch (Fig. 2), but in cells intra scratch we did not record the NCX current in the same experimental conditions;
  4. We used P/N leak subtraction to subtract the capacitive transient due to the patch pipette; 
  5. In the recording channel without P/N leak subtraction and in all experimental conditions tested, also after the NCX block, we didn’t record any difference in the capacitive transient.

It could be worth mentioning that we performed perforated patch-clamp recordings and no data are available, in our knowledge, on NCX recordings in cancer cells in CNS using this configuration. Most of the data in literature concerning the NCX current are instead recorded in whole-cell patch-clamp configuration and in heart myocytes, including the one cited by the reviewer (Hobai, Pflugers Arch. 433: 455–463 1997), describing NCX in rabbit ventricular myocytes. In myocytes the calcium dynamic could be different from CNS cancer cells, as the calcium dynamics are known to be profoundly altered as compared to other cells (Stewart et al., 2015). Since perforated patch-clamp recordings preserve the native ionic calcium concentration inside the cells and the intracellular milieu, this could be the decisive element at the basis of the differences that we observed. It should be noted that this exchanger is strongly conditioned by the ionic concentrations inside and outside the cells, pH, phosphatases and kinases, multiple pumps, and ion channels. Therefore, considering the complexity of modulation, we are currently performing experiments on this matter to provide a conclusive explication for the atypical current/voltage relationship. We had added a sentence in the discussion to underline the unusual I/V relationship of the NCX in our experimental conditions in line 590-591.

One aspect we wish to underline is that we tested the activation of the current in a range of membrane potentials of biological interest. 

We thank the reviewer for the suggestion to test the effect of Nickel on this current. We found that Nickel significantly reduces the inward current (Fig. 7). This proves the reliability of the current suggesting the involvement of the NCX transporter, as correctly pointed out by the reviewer. Similarly, using a 0 free Na+ extracellular solution significantly reduced the amplitude of the putative NCX-mediated current (Fig. 6).

Hobai IA, Khananshvili D, Levi AJ. The peptide "FRCRCFa", dialysed intracellularly, inhibits the Na/Ca exchange in rabbit ventricular myocytes with high affinity. Pflugers Arch. 1997 Feb;433(4):455-63. doi: 10.1007/s004240050300. PMID: 9000424.

Stewart TA, Yapa KT, Monteith GR. Altered calcium signaling in cancer cells. Biochim Biophys Acta. 2015 Oct;1848(10 Pt B):2502-11. doi: 10.1016/j.bbamem.2014.08.016. Epub 2014 Aug 20. PMID: 25150047.

Minor points

  1. For the level of the Ms, the title should be more simple and less existentialist. Something like ‘Role of Na+/Ca2+exchanger NCX in glioblastoma cell migration (in vitro)’ would look more appropriate.

We thank the reviewer for the suggestion. We modified the title following the reviewer's suggestion.

  1. P. 318-320. “After ouabain, the I/V relationship was shifted to the right of about 20 mV (insert in Figure 8)”. It is difficult to get how this figure (20 mV) has come out by looking at the inset.  

We thank the reviewer for pointing out the inaccuracy.  We have rephrased the sentence accordingly: 

Line 398: Following 500 µM ouabain perfusion, the I/V relationship shifted from inward to outward from the most hyperpolarized steps to +80mV (see also insert in Figure 10)

Line 402: “The insert of Figure 10 shows the averaged I/V magnified in the interval range between -40 mV to +40 mV to make more visible the I/V shift from inward to outward after the local perfusion of ouabain.”

  1. Conclusion. Conclusion is too wordy and way too much speculative. The weakness of many of the results allows neither of them. My suggestion is to halve the discussion and refrain from any urge of speculation.

We thank the reviewer for the suggestion. Accordingly, we removed all speculations from discussion section. 

Comments on the Quality of English Language

The English needs to be improved. 

We thank the reviewer for the point. Accordingly, we integrated and collected all the feedbacks from a mother-language colleague.

Reviewer 2 Report

The manuscript “ijms-2443782” provided some new information regarding the role of NCX in migration of glioblastoma cells. Although interesting, the data presentation raised some concerns:

1.      In abstract, “the natrium calcium exchanger (NCX)” is not a native English expression.

2.      The NCX is an exchanger. Whether it can also be called “pump”? Please justify it.

3.      Figure 8 shows “Effect of Na/K ATPase blocker on the NCX exchanger current”. However, the recording traces in Figure 8 are from Ca dependent BK channels, as in indicated in main text lines 316-318. This recording data needs to be revised.

4.      To study the action of bepridil on U251 cell migration, the author used bepridil at 50 µM. This concentration is a toxic dose for viability of glioma cells. They need to test bepridil at 10-20 µM.  

5.      In this study, only one NCX blocker was tested to show the role of NCX in glioma cell migration. To make a convincing conclusion, it is suggested to test other more selective NCX blockers on migration of glioma cells.

6.      Lines 446-447,  … the functional role of NCX in cancer are missing or scare [27]. Actually, there are many studies reported NCX actions in tumor cells. Please search literature in this research area and add more related information.

7.      Lines 436-437 are incomplete sentences. Please modify them.

Lines 436-437 are incomplete sentences. Please modify them.

Author Response

Comments and Suggestions for Authors

The manuscript “ijms-2443782” provided some new information regarding the role of NCX in migration of glioblastoma cells. Although interesting, the data presentation raised some concerns:

  1. In abstract, “the natrium calcium exchanger (NCX)” is not a native English expression.

We thank the reviewer for the point. We corrected it in sodium-calcium exchanger (NCX) through all the manuscript.

  1. The NCX is an exchanger. Whether it can also be called “pump”? Please justify it.

We thank the reviewer for pointing out the inaccuracy. We replaced “pump” with “exchanger” in the text. 

  1. Figure 8 shows “Effect of Na/K ATPase blocker on the NCX exchanger current”. However, the recording traces in Figure 8 are from Ca dependent BK channels, as in indicated in main text lines 316-318. This recording data needs to be revised.

We thank the reviewer for the point. We corrected the title in Figure 8 (actually Figure 10).

  1. To study the action of bepridil on U251 cell migration, the author used bepridil at 50 µM. This concentration is a toxic dose for viability of glioma cells. They need to test bepridil at 10-20 µM.  

We thank the reviewer for raising this issue. It has to be mentioned that several investigations proved that bepridil is able to induce cytotoxicity in different human tumor cell lines in a dose-dependent manner (Lee et al., 1995; Liu et al. 2022). Furthermore, recent data revealed the dose-dependent cytotoxicity of bepridil on both U87 and U251 cells leading to significant cell reduction. Specifically, U87 and U251 cell viability was reduced to 30-40%, 6-7%, and 3-4% of control level by 20‐, 50‐, and 100 μM bepridil. In particular, the concentrations of 18.6±1.1 and 14.9±1.1 µM were demonstrated to be a half maximal inhibitory concentration, able to induce cell death, causing a significant decrease (about 50%) in the number of U87 and U251 living proliferating cells, respectively (Hu et al., 2019). Nonetheless, it has to be underlined that all these findings, showing bepridil toxicity, referred to prolonged time of exposure, i.e. 48 and 72 hours. 

Concerning our data, it has to be taken into careful consideration that the exposure time to bepridil lasted 24 hours, therefore the treatment duration was shorter than those reported in previous studies. Additionally, it has to be highlighted that the bepridil dose employed in our current study was selected based on MTT vitality assay results. In particular, our findings obtained using U251 cells after 24 hours exposure to increasing concentrations of bepridil (20, 50, 70 and 100 μM), revealed that, interestingly, the tested dose of 20 μM unaffected cell viability, which persisted about 100%. Indicating a dose dependent cytotoxicity, the higher concentration of 50 μM was able to cause 30% cell viability decrease, while the concentration required to produce half maximal inhibition (IC50) of cell viability was about 60 μM. Therefore, having clear in mind the goal to clarify the role of NCX in U251 cell migration/invasiveness, we decided to chose the subtoxic concentration of 50 μM, leading to a mild reduction in the number of living/proliferation cells, as the proper suitable concentration to be used in the following experiments. 

We accordingly modified the text, to better explicit the choice of the employed bepridil concentration (see Results section). 

Lee YS, Sayeed MM, Wurster RD. Intracellular Ca2+ mediates the cytotoxicity induced by bepridil and benzamil in human brain tumor cells. Cancer Lett. 1995 6;88:87-91. doi: 10.1016/0304-3835(94)03619-t.

Liu, Z.; Cheng, Q.; Ma, X.; Song, M. Suppressing Effect of Na+/Ca2+ Exchanger (NCX) Inhibitors on the Growth of Melanoma Cells. Int. J. Mol. Sci. 2022, 23, 901. https://doi.org/10.3390/ ijms23020901. 

Hu HJ, Wang SS, Wang YX, Liu Y, Feng XM, Shen Y, Zhu L, Chen HZ, Song M. Blockade of the forward Na+/Ca2+ exchanger suppresses the growth of glioblastoma cells through Ca2+-mediated cell death. Br J Pharmacol. 2019 Aug;176(15):2691-2707. doi: 10.1111/bph.14692. 

  1. In this study, only one NCX blocker was tested to show the role of NCX in glioma cell migration. To make a convincing conclusion, it is suggested to test other more selective NCX blockers on migration of glioma cells.

We thank the reviewer for the point. We performed a new time lapse using Nickel, a known NCX blocker (Fig. 12). 

  1. Lines 446-447,  … the functional role of NCX in cancer are missing or scare [27]. Actually, there are many studies reported NCX actions in tumor cells. Please search literature in this research area and add more related information.

We thank the reviewer for the point. We changed the sentence as follow: “Several studies described the involvement of NCX in cardiac diseases and its inhibitors have been considered as therapeutic potential in hearth injury [26]. In contrast, the functional role of NCX in glioblastoma is not fully unveiled [27]. Indeed, in glioblastoma NCX has been reported to play a role mainly in proliferation and tumor growth [28,29], but little is known about its role in GBM cell.”

  1. Lines 436-437 are incomplete sentences. Please modify them.

We thank the reviewer for the point. We completed the sentences.

Comments on the Quality of English Language

Lines 436-437 are incomplete sentences. Please modify them.

Done.

Reviewer 3 Report

This is a very interesting paper exploring the natrium calcium exchanger (NCX) pump role in GBM invasiveness, and how its blockade could constitute a (probably adjuvant) therapeutic target. Adjuvant, since this is a relevant part but not the only challenge related to GBM. In any case, it is a well structured work with a carefully planned methodological approach. I only have minor comments/suggestions/corrections:

1) lines 418-419 and 482 I am unsure to understand author’s point here, since GBM in particular and brain cancer in general are not prone to generate metastasis. Invasion is indeed a relevant feature but not metastasis. If authors have different information that supports this claim, please provide references and % of metastasis originating from GBM

2) MTT is known to produce sometimes confusing results regarding viability. This is not a big issue inn the context of this paper anyway, but did authors confirmed viability with some other different methodological approach such as trypan blue?

3) Did authors check (or have literature data supporting) the % of stem cells present in U251 or U87?

4) Was integrity of the stablished cell lines checked? Especially U87 is known to have suffered a genetic drift depending on the lab and due to different conditions and the practice of “sharing” cell lines not checked. E.g. the NCI requires periodical integrity checking for the provided cell lines in order to renew MTAs.

Minor issues:

line 437 Starting of the sentence missing

line 501 why this reference citation is in a different format? 

English language quality is essentially fine, just take care of the following points:

line 428 ‘One of the strategies’ instead of ‘one of the strategy’

line 447 “scientific evidence… IS missing” not “are missing”

Author Response

Comments and Suggestions for Authors

This is a very interesting paper exploring the natrium calcium exchanger (NCX) pump role in GBM invasiveness, and how its blockade could constitute a (probably adjuvant) therapeutic target. Adjuvant, since this is a relevant part but not the only challenge related to GBM. In any case, it is a well structured work with a carefully planned methodological approach. I only have minor comments/suggestions/corrections:

1) lines 418-419 and 482 I am unsure to understand author’s point here, since GBM in particular and brain cancer in general are not prone to generate metastasis. Invasion is indeed a relevant feature but not metastasis. If authors have different information that supports this claim, please provide references and % of metastasis originating from GBM

We thank the reviewer for pointing out the inaccuracy. We removed “metastasis” and all related concept in the text.

2) MTT is known to produce sometimes confusing results regarding viability. This is not a big issue inn the context of this paper anyway, but did authors confirmed viability with some other different methodological approach such as trypan blue?

We thank the reviewer for raising this issue. The MTT is the most widely colorimetric test originally developed to measure the metabolic activity of live cells. Over the years, the reduction of MTT (3-(4, 5-dimethylthiazolyl-2)-2, 5-diphenyltetrazolium bromide) by live cells in water-insoluble violet-blue molecules called formazan salts has been widely accepted as a reliable assay for measuring cell proliferation in vitro (Berridge et al., 2005, Talorete et al., 2006).

We are aware that, despite the numerous advantages offered by this technique, some limitations and criticism must be considered in the execution and interpretation of MTT assay (Wang et al., 2010; Aslantürk, 2017; Ghasemi et al., 2021), therefore we took several good laboratory practices and precautions.

It should be noted that, in order to exclude the cytotoxicity of bepridil and therefore establish the optimal use concentration for subsequent experiments, we here applied a well-optimized protocol already used on the U251 cell line in the laboratory (Ferrari et al., 2021).

The first critical point to take into consideration when designing the MTT assay is the number of cell-seeded/ well to guarantee reproducible results. Indeed, a discrepancy in the number of seeded cells could vary the amount of formazan salts formed and the optical density measured. Furthermore, two other important aspects to consider are the growth rate and the cell density in the wells. Indeed, cells in over-confluence could be influenced by excessive spatial proximity to other cells and thus alter their metabolic profile. All together these variations could give rise to inconsistency in the results (Ghasemi et al., 2021).

In the current study a standardized, reproducible protocol was applied, coming from long-lasting years’ experience and improvements over time. Concerning the MTT data reported in Figure 13, our in-use protocol was applied considering both the well surface area and the growth rate of U251 glioblastoma cells. In detail, we seeded the same volume of cells (about 10.000 cells/well) reaching 80% of confluence after 24 h. This allows us to obtain consistent results avoiding signal fluctuation and/or saturation. 

In addition, to avoid any possible cytotoxic effect of MTT, signal saturation and/or false-positive results, we used freshly MTT solution, and we applied standardized MTT concentrations and incubation time based on our previous studies using different cell lines evaluating diverse compounds (Ferrari et al., 2021; Gola et al., 2023; Coccini et al., 2010). Furthermore, after solubilization with high grade DMSO, we immediately read the plate with spectrophotometer avoiding light exposure to ensure the correct reading of the optical signal without interference. 

Trypan blue is a dye exclusion method that elicits to evaluate the cellular membrane integrity. However, with this assay, cells that are still alive but losing their function will not be stained. Despite its easy to perform characteristics, it is challenging for the experimental practice of bulky sample sizes with differences in the timing of progressive cytotoxic effects as is the case of bepridil. Moreover, counting errors may occur, particularly when using the assay in absence of an automatic cell counter, therefore running into operator influence and manual error.

Indeed, thanks to MTT assay results, we can assume that the treatments induced cytotoxicity and cell death after 24 hours of incubation. In addition, trypan blue does not allow to discriminate between healthy cells and/or cells that slow their metabolism giving rise to an erroneous estimation of cells viability (Mani and Swargiary, 2023). Most importantly, trypan blue is suitable for suspended cell lines rather than monolayer cells since the latter must be trypsinized, resuspended in the dye and plated before performing the test. Lastly, it was demonstrated that trypan blue is toxic for mammalian cells. Therefore, exposing the cells to trypan blue for a long time can cause cytotoxicity and reduce the number of viable cells.

Overall, it has also to be highlighted that in the current study, the MTT assay is not the central topic in our mind, but it should be considered in the context of the whole investigation as a tool to assess the proper bepridil concentration to be used in the following experimental determination.

Aslantürk ÖS. In Vitro Cytotoxicity and Cell Viability Assays: Principles, Advantages, and Disadvantages. Genotoxicity - A Predictable Risk to Our Actual World. InTech; 2018. Available from: http://dx.doi.org/10.5772/intechopen.71923

Berridge MV, Herst PM, Tan AS. Tetrazolium dyes as tools in cell biology: new insights into their cellular reduction. Biotechnol Annu Rev. 2005; 11:127-52. doi: 10.1016/S1387-2656(05)11004-7. PMID: 16216776.

Coccini T, Roda E, Sarigiannis DA, Mustarelli P, Quartarone E, Profumo A, Manzo L. Effects of water-soluble functionalized multi-walled carbon nanotubes examined by different cytotoxicity methods in human astrocyte D384 and lung A549 cells. Toxicology. 2010 Feb 28;269(1):41-53. doi: 10.1016/j.tox.2010.01.005. Epub 2010 Jan 14. PMID: 20079395.

Ferrari B, Roda E, Priori EC, De Luca F, Facoetti A, Ravera M, Brandalise F, Locatelli CA, Rossi P, Bottone MG. A New Platinum-Based Prodrug Candidate for Chemotherapy and Its Synergistic Effect With Hadrontherapy: Novel Strategy to Treat Glioblastoma. Front Neurosci. 2021 Mar 22; 15:589906. doi: 10.3389/fnins.2021.589906. PMID: 33828444; PMCID: PMC8019820.

Ghasemi M, Turnbull T, Sebastian S, Kempson I. The MTT Assay: Utility, Limitations, Pitfalls, and Interpretation in Bulk and Single-Cell Analysis. Int J Mol Sci. 2021 Nov 26;22(23):12827. doi: 10.3390/ijms222312827. PMID: 34884632; PMCID: PMC8657538.

Gola F, Gaiaschi L, Roda E, De Luca F, Ferulli F, Vicini R, Rossi P, Bottone MG. Voghera Sweet Pepper: A Potential Ally against Oxidative Stress and Aging. Int J Mol Sci. 2023 Feb 14;24(4):3782. doi: 10.3390/ijms24043782. PMID: 36835192; PMCID: PMC9959306.

Mani, S., Swargiary, G. (2023). In Vitro Cytotoxicity Analysis: MTT/XTT, Trypan Blue Exclusion. In: Animal Cell Culture: Principles and Practice. Techniques in Life Science and Biomedicine for the Non-Expert. Springer, Cham. https://doi.org/10.1007/978-3-031-19485-6_18

Mani, S., Swargiary, G. (2023). In Vitro Cytotoxicity Analysis: MTT/XTT, Trypan Blue Exclusion. In: Animal Cell Culture: Principles and Practice. Techniques in Life Science and Biomedicine for the Non-Expert. Springer, Cham. https://doi.org/10.1007/978-3-031-19485-6_18.

Talorete TP, Bouaziz M, Sayadi S, Isoda H. Influence of medium type and serum on MTT reduction by flavonoids in the absence of cells. Cytotechnology. 2006 Nov;52(3):189-98. doi: 10.1007/s10616-007-9057-4. Epub 2007 Feb 24. PMID: 19002877; PMCID: PMC3449413.

Wang P, Henning SM, Heber D. Limitations of MTT and MTS-based assays for measurement of antiproliferative activity of green tea polyphenols. PLoS One. 2010 Apr 16;5(4):e10202. doi: 10.1371/journal.pone.0010202. PMID: 20419137; PMCID: PMC2855713.

3) Did authors check (or have literature data supporting) the % of stem cells present in U251 or U87?

We thank the reviewer for raising this issue. Literature data reported that U251 cells were found to contain about 3% CD133+ cells, whereas U87 cells contained only about 1% CD133+ cells (Jung et al., 2013), where CD133 has been used as a marker to identify glioma stem cells (Glumac et al., 2018). However, there are emerging contradictions, and CD133 glioma stem cells exist (Zhang et al., 2013). So, we can speculate that the number of stem cells in U251 and U87 could be higher than previously indicated. Indeed, other markers, such as Sox2, induce glioblastoma cell stemness (Lopez-Bertoni et al., 2022). In particular, a study conducted by Wen and colleagues (2023) demonstrated that SOX2 expression could be detected in greater than 70% of U251 cells, but less than 20% of U87 cells. 

Glumac PM, LeBeau AM. The role of CD133 in cancer: a concise review. Clin Transl Med. 2018 Jul 9;7(1):18. doi: 10.1186/s40169-018-0198-1. PMID: 29984391; PMCID: PMC6035906.

Jung TY, Choi YD, Kim YH, Lee JJ, Kim HS, Kim JS, Kim SK, Jung S, Cho D. Immunological characterization of glioblastoma cells for immunotherapy. Anticancer Res. 2013 Jun;33(6):2525-33. PMID: 23749904.

Zhang S, Xie R, Wan F, Ye F, Guo D, Lei T. Identification of U251 glioma stem cells and their heterogeneous stem-like phenotypes. Oncol Lett. 2013 Dec;6(6):1649-1655. doi: 10.3892/ol.2013.1623. Epub 2013 Oct 11. PMID: 24260059; PMCID: PMC3834304.

Lopez-Bertoni H, Johnson A, Rui Y, Lal B, Sall S, Malloy M, Coulter JB, Lugo-Fagundo M, Shudir S, Khela H, Caputo C, Green JJ, Laterra J. Sox2 induces glioblastoma cell stemness and tumor propagation by repressing TET2 and deregulating 5hmC and 5mC DNA modifications. Signal Transduct Target Ther. 2022 Feb 9;7(1):37. doi: 10.1038/s41392-021-00857-0. PMID: 35136034; PMCID: PMC8826438.

Wen L, Wang XZ, Qiu Y, Zhou YP, Zhang QY, Cheng S, Sun JY, Jiang XJ, Rayner S, Britt WJ, Chen J, Hu F, Li FC, Luo MH, Cheng H. SOX2 downregulation of PML increases HCMV gene expression and growth of glioma cells. PLoS Pathog. 2023 Apr 14;19(4):e1011316. doi: 10.1371/journal.ppat.1011316. PMID: 37058447; PMCID: PMC10104302.

4) Was integrity of the stablished cell lines checked? Especially U87 is known to have suffered a genetic drift depending on the lab and due to different conditions and the practice of “sharing” cell lines not checked. E.g. the NCI requires periodical integrity checking for the provided cell lines in order to renew MTAs.

We thank the reviewer for the point. We are agreeing that the U87 cell line is known to have suffered a genetic drift depending on the lab and due to different conditions. For this reason, in this paper we used 3 different cell lines for obtaining the most possible identification data of GBM. Finally, we added in the “Acknowledges” section the origin of the U87 cell line.

Minor issues:

line 437 Starting of the sentence missing

Done

line 501 why this reference citation is in a different format? 

We thank the reviewer for pointing out the inaccuracy. We corrected the citation format.

Comments on the Quality of English Language

English language quality is essentially fine, just take care of the following points:

line 428 ‘One of the strategies’ instead of ‘one of the strategy’

Done.

line 447 “scientific evidence… IS missing” not “are missing”

Done.

Round 2

Reviewer 1 Report

Unfortunately, I'm still unconvinced about the nature of the current recorded by the authors as an NCX. Their words worried me even more about the possibility that the recorded current is an artifact. They are indeed using a P/N protocol that could actually give rise to the artifact.

In any case, it is very hard for me to understand how the conditions of a perforated patch configuration could allow an inward current at a membrane potential of +140 mV (cf their Fig 4).

The authors shoud isolate the NCX current without a P/N protocol, performing subtraction between ctrl and Ni conditions. I would expect a very different shape of the NCX IV relationship obtained using this method

To be improved

Reviewer 2 Report

The manuscript has been well revised.

Minor editing of English language required.